# Dynamic representation of 3D auditory space in the midbrain of the free-flying echolocating bat

**Ninad B Kothari[†], Melville J Wohlgemuth[†], Cynthia F Moss***

Johns Hopkins University, Baltimore, United States

**Abstract** Essential to spatial orientation in the natural environment is a dynamic representation of direction and distance to objects. Despite the importance of 3D spatial localization to parse objects in the environment and to guide movement, most neurophysiological investigations of sensory mapping have been limited to studies of restrained subjects, tested with 2D, artificial stimuli. Here, we show for the first time that sensory neurons in the midbrain superior colliculus (SC) of the free-flying echolocating bat encode 3D egocentric space, and that the bat's inspection of objects in the physical environment sharpens tuning of single neurons, and shifts peak responses to represent closer distances. These findings emerged from wireless neural recordings in free-flying bats, in combination with an echo model that computes the animal's instantaneous stimulus space. Our research reveals dynamic 3D space coding in a freely moving mammal engaged in a real-world navigation task.

DOI: https://doi.org/10.7554/eLife.29053.001

## Introduction

As humans and other animals move in a 3D world, they rely on dynamic sensory information to guide their actions, seek food, track targets and steer around obstacles. Such natural behaviors invoke feedback between sensory space representation, attention and action-selection (*Lewicki et al., 2014*). Current knowledge of the brain's representation of sensory space comes largely from research on neural activity in restrained animals, generally studied with 2D stimuli (*Van Horn et al., 2013*); however, far less is known about 3D sensory representation, particularly in freely moving animals that must process changing stimulus information to localize objects and guide motor decisions as they navigate the physical world.

Animals that rely on active sensing provide a powerful system to investigate the neural underpinnings of sensory-guided behaviors, as they produce the very signals that inform motor actions. Echolocating bats, for example, transmit sonar signals and process auditory information carried by returning echoes to guide behavioral decisions for spatial orientation (*Griffin, 1958*). Work over the past decade has revealed that echolocating bats produce clusters of sonar calls, termed sonar sound groups (SSGs), to closely inspect objects in their surroundings or to negotiate complex environments (*Kothari et al., 2014*; *Moss et al., 2006*; *Petrites et al., 2009*; *Sändig et al., 2014*). We hypothesize that the bat's sonar inspection behavior sharpens spatio-temporal echo information processed by the auditory system in a manner analogous to the active control of eye movements to increase visual resolution through sequences of foveal fixations (*Hayhoe and Ballard, 2005*; *Moss and Surlykke, 2010*; *Tatler et al., 2011*). Importantly, the bat's acoustic behaviors provide a quantitative metric of spatial gaze, and can thus be analyzed together with neural recordings to investigate the dynamic representation of sensory space.

Echolocating bats compute the direction of echo sources using a standard mammalian auditory system (*Wohlgemuth et al., 2016*). The dimension of target distance is computed from the time

**\*For correspondence:**
cynthia.moss@jhu.edu

[†]These authors contributed equally to this work

**Competing interests:** The authors declare that no competing interests exist.

**eLife digest** Humans and other animals can navigate their natural environments seamlessly, even if there are obstacles in their path. However, it is not well understood how an animal's brain processes information from the senses to map where it is in relation to these objects, both in terms of distance and direction.

Bats can help answer these questions because they use a biological navigation system: echolocation. Bat produce high-pitched squeaks and then listen to the echoes that return when the sound bounces off of nearby objects. A bat can then use this information to estimate both which direction an object is, and how far away it is. Bats can also change their echolocation signals to focus on different objects, and researchers can record and analyze these signals to directly measure what the bat is paying attention to.

Kothari, Wohlgemuth and Moss have now investigated how the brain cells of bats process the animals' movements while flying in three-dimensional space. A wireless probe was inserted into the midbrain region of each bat to detect whenever there was an electrical impulse in the nearby brain cells. The bats were then allowed to fly freely in a large room that contained obstacles, while each bat's echolocation signals and brain activity were recorded.

The experiments revealed a group of brain cells that codes for the position of an object in three-dimensional space. Kothari, Wohlgemuth and Moss noted that these brain cells register the distance to objects more precisely when the bat changed its echolocation behavior to focus on those objects. Moreover, the activity in the bat's brain also shifted when the bat noticed a closer object. These findings are not only relevant to echolocating bats, but rather reflect the general role that shifts in attention may play when many species map the locations of objects around them.

Further similar studies with other species would contribute to a more complete understanding of animals' nervous systems work under natural conditions. In the future, these findings, and the studies that build upon them, could be applied to other fields of research like medicine or engineering. For example, smart wireless devices, designed to record and transmit physiological measurements based on movement, could efficiently monitor human health, and robots equipped with adaptive sonar could navigate effectively in complex environments.

DOI: https://doi.org/10.7554/eLife.29053.002

delay between sonar emissions and echoes (*Simmons, 1973*). Neurophysiological investigations of echo processing in bats reveal that a class of neurons shows facilitated and delay-tuned responses to **simulated** pulse-echo pairs. It has been hypothesized that echo delay-tuned neurons carry information about the distance to objects (*Feng et al., 1978*; *O''Neill and Suga, 1982*; *Suga and O'Neill, 1979*; *Valentine and Moss, 1997*); however, the neural representation of target distance in bats listening to self-generated echoes reflected from physical objects has never previously been empirically established.

The midbrain superior colliculus (SC) has been implicated in sensory-guided spatial orienting behaviors, such as visual and auditory gaze control in primates, cats and barn owls (*Knudsen, 1982*; *Krauzlis, 2004*; *du Lac and Knudsen, 1990*; *Middlebrooks and Knudsen, 1984*; *Munoz et al., 1991*; *Sparks, 1986*; *Stein et al., 1989*), prey-capture behavior in frog and pit viper (*Grobstein, 1988*; *Hartline et al., 1978*; *Newman and Hartline, 1981*), and echolocation in bats (*Valentine and Moss, 1997*; *Valentine et al., 2002*). Previous work has also demonstrated that the SC is an integral part of the *egocentric* spatial attention network, specifically for target selection and goal-directed action (*Krauzlis et al., 2013*; *Lovejoy and Krauzlis, 2010*; *McPeek and Keller, 2004*; *Mysore and Knudsen, 2011*; *Zénon and Krauzlis, 2012*). Work in freely behaving rodents has also demonstrated a more general role of the SC in sensory-guided orienting behaviors (*Duan et al., 2015*; *Felsen and Mainen, 2008*). Additionally, measures of the local field potential (LFP) in the midbrain optic tectum (avian homologue of the SC) have shown that increases in the gamma band (~40–140 Hz) correlate with attention to sensory stimuli (*Sridharan and Knudsen, 2015*). The research reported here is the first to investigate the behavioral modulation of depth-tuned single unit responses and gamma band oscillations in the SC of a mammal inspecting objects in its physical environment.

Prior work on sensorimotor representation in the mammalian SC has been largely carried out in restrained animals performing 2D tasks, leaving gaps in our knowledge about the influence of action and attention on sensory responses in animals moving freely in a 3D physical environment. To bridge this gap, we conducted wireless chronic neural recordings of both single unit activity and LFPs in the SC of free-flying bats that used echolocation to localize and inspect obstacles along their flight path. Central to this research, we developed a novel echo model to reconstruct the bat's instantaneous egocentric stimulus space, which we then used to analyze echo-evoked neural activity patterns. Our data provide the first demonstration that neurons in the midbrain SC of a *freely moving animal represent the 3D egocentric location of physical objects in the environment, and that active sonar inspection sharpens and shifts the depth tuning of 3D neurons.*

## Results

Big brown bats, *Eptesicus fuscus,* flew in a large experimental test room and navigated around obstacles (*Figure 1A*, wall landing; *Figure 1B*, platform landing); they received a food item after each landing. The bats showed natural adjustments in flight and sonar behaviors in response to echoes arriving at their ears from objects in the room. The positions of objects were varied across recording sessions, and the bats were released from different points in the room within recording sessions, to limit their use of spatial memory for navigation and instead invoke their use of echo feedback. We specifically tested whether the bats relied on spatial memory to guide their navigation by analyzing their flight trajectories over repeated trials. Our analysis considered whether the bats showed stereotypy in their flight paths, an indicator of memory-based flight (*Griffin, 1958*), by constructing 2D spatial cross correlations of the flight trajectories across trials within each experimental session (*Barchi et al., 2013*). Our results show low correlation numbers, and confirm that bats were not relying on spatial memory (*Falk et al., 2014*), but instead active sensing, in this flight task (*Figure 1—figure supplement 1*, also see Materials and methods).

While the bats performed natural sensory-guided behaviors, sonar calls were recorded using a wide-band ultrasound microphone array (*Figure 1A,B* – grey circles are microphones; see *Figure 1B*, raw oscillogram in middle panel and spectrograms in bottom panel and inset). The bat's 3D flight trajectory and head aim were measured using high-speed Vicon motion capture cameras (*Figure 1A,B*, frame-rate 300 Hz). In flight, bats displayed natural adaptations in sonar behavior (*Griffin, 1958*; *Simmons et al., 1979*). Specifically, they increased echolocation pulse rate (PR) and decreased pulse duration (PD) as they approached objects or their landing points (*Figure 1E*), and they also produced sonar sound groups (SSGs), or clusters of vocalizations, to inspect objects in space (*Falk et al., 2014*; *Moss et al., 2006*; *Petrites et al., 2009*; *Sändig et al., 2014*; *Wheeler et al., 2016*).

Extracellular neural activity was recorded with a 16-channel silicon probe, affixed to a microdrive, and implanted in the bat SC. Neural activity was transmitted wirelessly via radio telemetry (Triangle BioSystems International; *Figure 1A* – green box). *Figure 1C* shows histology of SC recording sites, and *Figure 1D* shows simultaneous neural recordings from two channels (see also Materials and methods). *Figure 1—figure supplement 2*, demonstrates single cell neural recordings across multiple trials (also see *Figure 1—figure supplement 3* for clustering efficacy).

### Echo model - Reconstructing the instantaneous acoustic stimulus space at the ears of the bat

To measure auditory spatial receptive fields in the bat SC, we first determined the azimuth, elevation and distance of objects, referenced to the bat's head direction and location in the environment (*Figure 2—figure supplement 1A* shows a cartoon of a bat with a telemetry recording device and markers to estimate the bat's head direction, *Figure 2—figure supplement 1B* shows a top view of the bat's head with the telemetry device and head tracking markers, also see Materials and methods). In order to determine the 3D direction and arrival time of sonar echoes returning to the bat, we relied on the physics of sound to establish an echo model of the bat's instantaneous sensory space. The echo model takes into account an estimate of the beam width of the bat's sonar calls, its 3D flight trajectory, its head direction, as well as physical parameters of sound (*Figure 2—figure supplement 1A and B* – schematic, see Materials and methods) to compute a precise estimate of the time of arrival of echoes at the bat's ears, as well as the 3D location of the echo sources

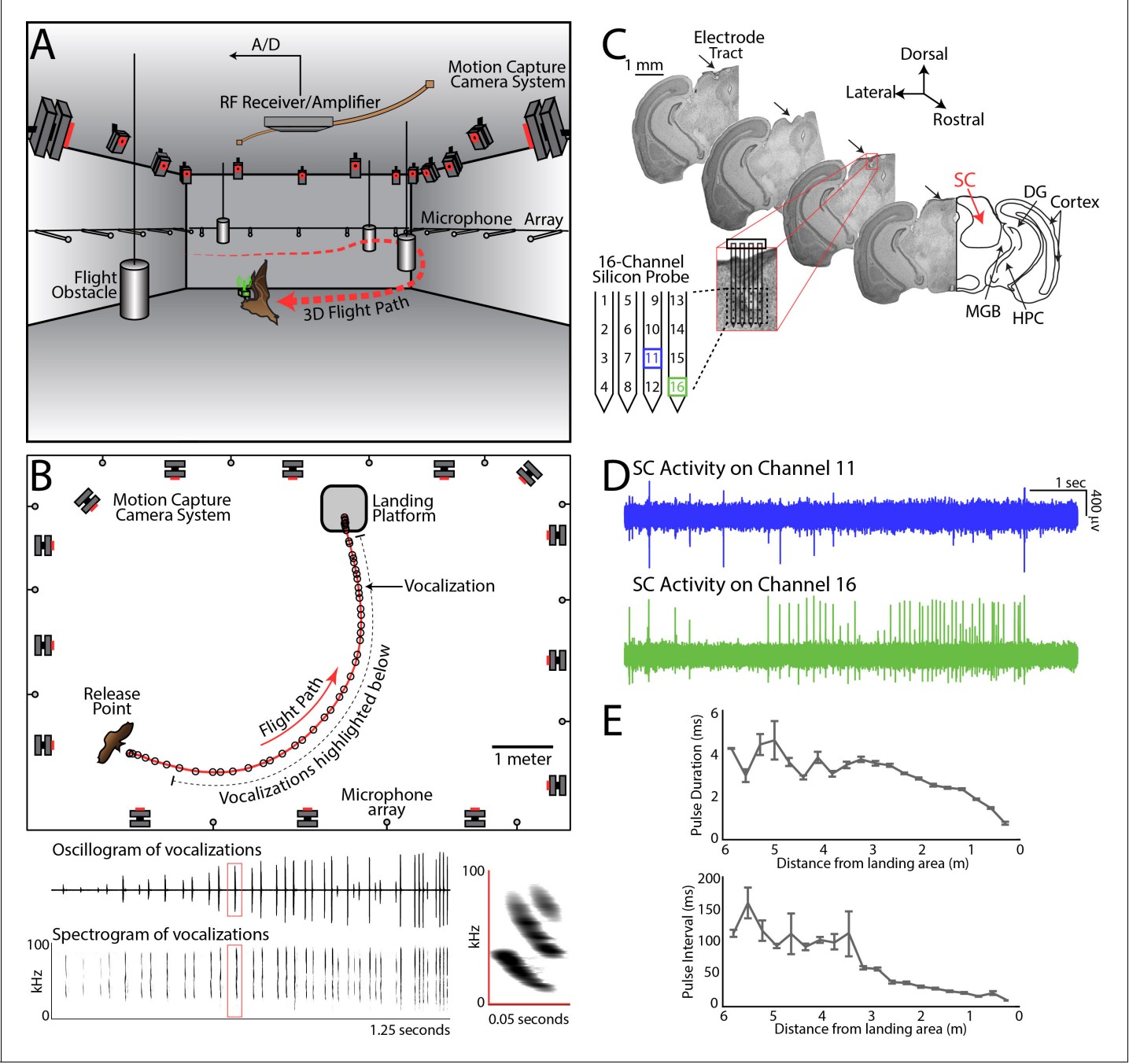

**Figure 1.** Experimental setup and methodology. (A) Configuration of the experimental flight room for wireless, chronic neural recordings from freely flying echolocating bats. Shown is the bat (in brown) with the neural telemetry device mounted on the head (in green). The telemetry device transmits RF signals to an RF receiver connected to an amplifier and an analog-to-digital recording system. The bat's flight path (in red) is reconstructed by 16 motion capture cameras (not all are shown) tracking three reflective markers mounted on the dorsal surface of the telemetry device (3 mm round hemispheres). While the bat flies, it encounters four different, cylindrical flight obstacles (shown in grey), and the sonar vocalizations are recorded with a wide-band microphone array mounted on the walls. (B) Overhead view of the room in the platform-landing task. The bat flew across the room (red line) using echolocation to navigate (black circles are sonar vocalizations) while recordings were made wirelessly from the SC (as shown in pane A). Vocalizations produced on this trial are shown in greater detail in bottom panels (filtered audio trace and corresponding spectrogram). The inset, on the right, shows a zoomed-in view of the spectrogram of one call, indicated by the red box. (C) Histological reconstruction of the silicon probe tract through the superior colliculus (SC) of one bat in the study. Shown are four serial coronal sections, approximately 2.5 mm from bregma, at the location of the SC. Lesions at the site of the silicon probe are indicated with black arrows. Also marked in the most rostral section are the locations of the SC, medial geniculate body (MGB), hippocampus (HPC), cortex, and dentate gyrus (DG). (D) Simultaneous neural recordings from SC from the recording

*Figure 1 continued on next page*

*Figure 1 continued*

sites identified with a blue square and green square in the silicon probe layout panel in *Figure 1C*. (layout of the 16-channel silicon probe used for SC recordings). (E) Top, change in sonar pulse duration as a function of object distance. Bottom, change in pulse interval as a function of object distance.

DOI: https://doi.org/10.7554/eLife.29053.003

The following figure supplements are available for figure 1:

**Figure supplement 1.** Cross-correlation of flight paths.
DOI: https://doi.org/10.7554/eLife.29053.004
**Figure supplement 2.** Spike waveform consistency throughout a single recording session.
DOI: https://doi.org/10.7554/eLife.29053.005
**Figure supplement 3.** Spike cluster separation.
DOI: https://doi.org/10.7554/eLife.29053.006

(*Figure 2A* – cartoon explains the echo model, with cones showing the sonar beam pattern, *Figure 2B* – the time series of call and echoes from the cartoon in *Figure 2A*; *Figure 2C* – actual bat flight trajectory with sonar vocalizations, orange circles, and 3D head aim vectors, black lines; *Figure 2D and E* – the instantaneous solid angles of the head aim with respect to objects and echo arrival times of sonar returns from different objects along the trajectory in 2C; also see Materials and methods).

The echo model was used to construct the instantaneous acoustic sensory space of the bat each time it vocalized and received echoes from physical objects in its flight path. We first determined the onset of each vocalization produced by the bat, then the 3D position of the bat at the time of each sonar vocalization, and the 3D relative positions of flight obstacles. Past work has demonstrated that the big brown bat's sonar beam axis is aligned with its head (*Ghose and Moss, 2003*; *2006*), and the direction of the sonar beam was inferred in our study from the head-mounted markers showing the head aim of the bat. We then referenced the 50 deg −6 dB width of the sonar beam at 30 kHz (*Hartley and Suthers, 1989*), and the time at which the sonar beam reflected echoes from flight obstacles in the animal's path. From this calculation, we computed the direction and time of arrival of all echoes returning to the bat's ears each time the animal emitted a sonar call.

Although it is possible to use a wireless, head-mounted microphone to record the returning echo stream, there are significant limitations to this methodology. First, a single head-mounted microphone has a higher noise floor than the bat's auditory receiver and therefore does not pick up all returning echoes that the bat may hear. Moreover, a single microphone would add weight to the devices carried by the bat in flight and could only provide information regarding echo arrival time, not sound source direction. A head-mounted microphone is therefore insufficient to compute the 3D locations of echo sources, thus highlighting the importance of the echo model in our study to compute the bat's instantaneous 3D sensory space.

We computed errors in the measurements of head-aim as well as in the estimation of echo arrival times at the bat's ears (*Figure 2—figure supplement 2*). Our measurements indicate that the maximum error in the reconstruction of the bat head-aim does not exceed 5.5 degrees, and the error in echo arrival time measurement is between 0.35 and 0.65 ms (see *Figure 2—figure supplement 2C and D* – estimation of errors in head-aim reconstruction, *Figure 2—figure supplement 2* – errors in echo arrival time; see Materials and methods). To confirm that the echo model accurately calculated the 3D positions of sonar objects, we used echo playbacks from a speaker and microphone pair (see Materials and methods, *Figure 2—figure supplement 2*), with additional validation by using a microphone array placed behind the bat's flight direction. The microphone array recorded the echoes reflected off objects as the bat flew and produced sonar vocalizations, which were analyzed with time of arrival difference (TOAD) algorithms to compare the measured echo sources with the calculated echo sources based on our echo model (see Materials and methods).

## 3D spatial tuning of single neurons in the SC of free flying bats

The establishment of the echo model was a critical step in computing 3D spatial tuning of SC neurons recorded from the animals in flight. The spatial acoustic information (echo arrival times and 3D locations of echo sources) obtained from the echo model was converted into 3D egocentric coordinates to compute the acoustic stimulus space from the point of view of the flying bat as it navigated the room (*Figure 1—figure supplement 1F and G*, see Materials and methods). Bats were released

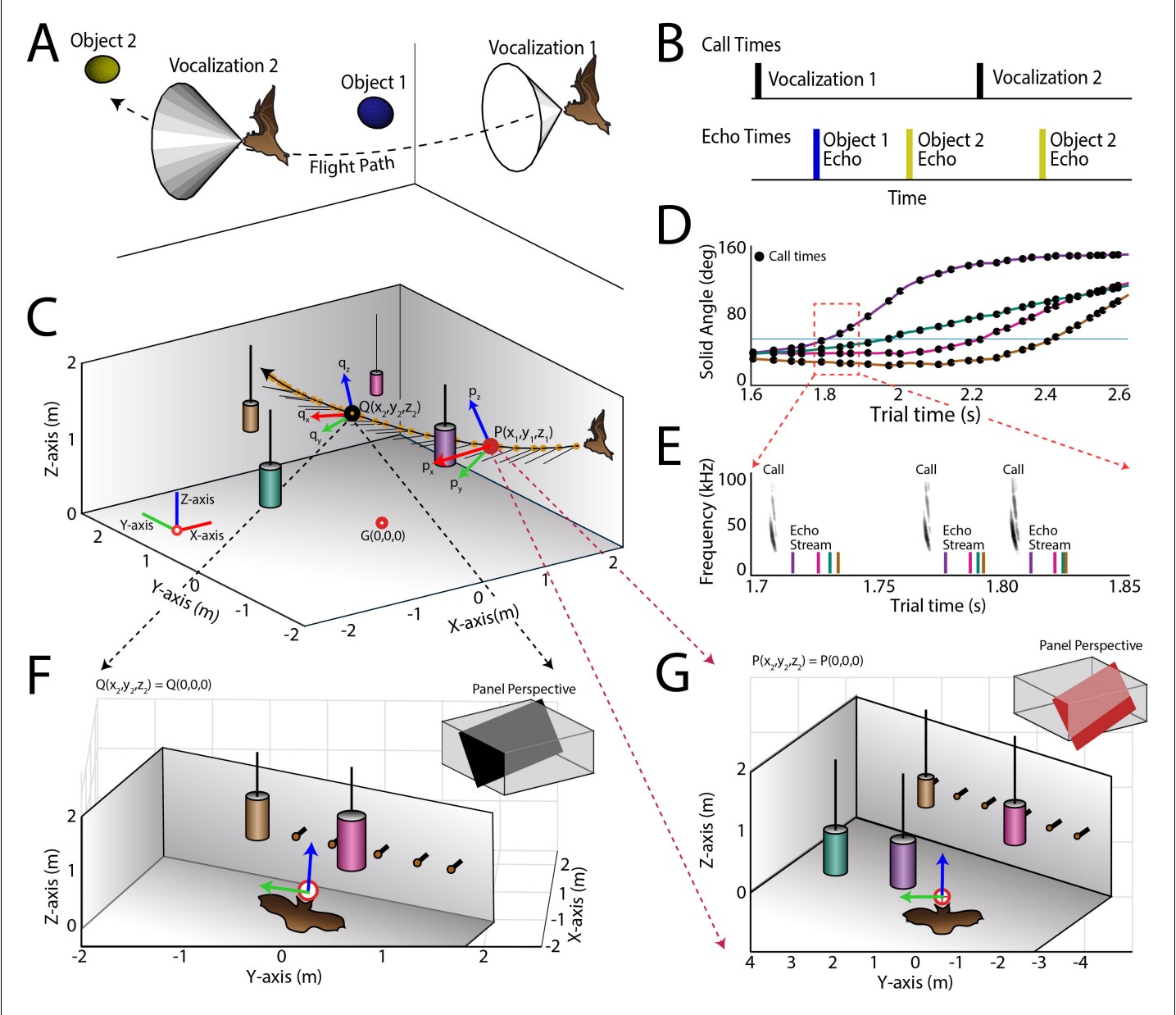

**Figure 2.** Use of the echo model to determine the bat's ongoing sensory signal reception. (A) Cartoon of a bat flying through space encountering two obstacles. The bat's flight trajectory moves from right to left, and is indicated by the black dotted line. Two sonar vocalizations while flying are indicated by the gray cones. (B) Reconstruction of sonar vocal times (top), and returning echo times (bottom) for the cartoon bat in panel a. Note that two echoes (blue and yellow) return to the bat following the first sonar vocalization, while only one echo (yellow) returns after the second vocalization, because the relative positions of the bat and objects change over time. (C) One experimental trial of the bat flying and navigating around obstacles (large circular objects). The bat's flight path (long black line) starts at the right and the bat flies to the left. Each vocalization is indicated with a yellow circle, and the direction of the vocalization is shown with a short black line. (D) Trial time versus solid angle to each obstacle for flight shown in C. Individual vocalizations are indicated with black circles, and the color of each line corresponds to the objects shown in C. (E) Time expanded spectrogram of highlighted region in D. Shown are three sonar vocalizations, and the colored lines indicate the time of arrival of each object's echo as determined by the echo model (colors as in C and D). (F) Snapshot of highlighted region (open black circle) in panel C showing the position of objects when the bat vocalized at that moment. (G) Snapshot of highlighted region (open red circle) in panel C showing the position of objects when the bat vocalized at that moment. In panels F and G, orange circles are microphones (only part of the array is shown here).

DOI: https://doi.org/10.7554/eLife.29053.007

The following figure supplements are available for figure 2:

**Figure supplement 1.** Head aim reconstruction.

DOI: https://doi.org/10.7554/eLife.29053.008

*Figure 2 continued on next page*

*Figure 2 continued*

**Figure supplement 2.** Error analysis and validation of the echo-model.
DOI: https://doi.org/10.7554/eLife.29053.009

from different locations in order to cover the calibrated volume of the flight room (*Figure 3—figure supplement 1A*), and they continuously produced echolocation calls, which resulted in series of echoes from objects during each recording session (*Figure 3—figure supplement 1A,B and C*). We also released the bats from multiple locations in the room so that they took a variety of flight paths through the room, and interacted with the flight obstacles from a broad range of directions and distances, which is necessary for computing spatial receptive fields. These data therefore yielded measurements of echoes returning to the animal from objects at many different directions and distances in egocentric space (*Figure 3—figure supplement 1D* - range coverage, E - azimuth coverage, and F - elevation coverage).

The output of the echo model was used to analyze audio/video-synchronized neural recordings from single units (see *Figure 1E*, *Figure 1—figure supplement 1* and Materials and methods) taken in the midbrain SC using a 16-channel wireless telemetry system. We recorded a total of 182 single neurons. We then classified neurons as sensory (n = 67), sensorimotor (45), vocal premotor (n = 26), or unclassified (n = 44), as described in the Materials and methods section. Here we focus on sensory neurons in the SC of free-flying bats.

For all sensory neurons we first calculated the distance, or echo-delay tuning (*Figure 3A and B*). An example reconstruction of a neuron's spatial tuning along the distance axis is displayed in *Figure 3B*, showing neural activity aligned to sonar vocalization times (red arrows), and responses to echoes returning at ~10 ms delay. Arrival time of the first echo at the bat's ears is indicated with a green arrow, and a second returning echo (from another, more distant object) is indicated with a blue arrow. Note that this example neuron does not spike in response to the second echo, nor to echoes arriving very early (*Figure 3C*, top panel), or late (*Figure 3C*, bottom panel). *Figure 3D* shows the computed distance (echo-delay) tuning profile of this same example neuron.

Using the echo model, we also calculated the tuning profiles of each neuron in azimuth and elevation (*Figure 3—figure supplement 2A* – azimuth and B – elevation). Once we calculated the azimuth, elevation, and distance tuning of neurons (*Figure 4A*), we constructed three-dimensional spatial response profiles for each neuron. *Figure 4B* shows surface plots of the three-dimensional tuning for two other example neurons. Of the 67 single sensory neurons (Bat 1–28 in green, and Bat 2–39 in brown) recorded in the SC of two big brown bats, 46 neurons (Bat 1–19 and Bat 2–27) in the data set showed selectivity to stimulus locations in 3D egocentric space (*Figure 4C*, see Materials and methods for details about spatial selectivity analysis), and these spatial tuning profiles were stable within recording sessions (*Figure 4—figure supplement 1*). Additionally, the selectivity of the neurons, in the distance dimension, did not vary as a function of dorsal-ventral location in the SC (*Figure 4—figure supplement 2*). Further, three neurons were tuned to both azimuth and range, two were tuned to both range and elevation, and five, three and three neurons were tuned exclusively to range, azimuth and elevation, respectively (see *Figure 4—figure supplement 3*). Best echo delays spanned values of 4 to 12 ms, corresponding to the distances of objects encountered by the bat (~70–200 cm) in our flight room (*Figure 4D, E and F* show histograms of standard deviations of normal fits to spatial receptive fields, also see Materials and methods).

## Adaptive sonar behavior modulates 3D spatial receptive fields

Guided by growing evidence that an animal's adaptive behaviors and/or attentional state can modulate sensory responses of neurons in the central nervous system (*Bezdudnaya and Castro-Alamancos, 2014*; *Fanselow and Nicolelis, 1999*; *McAdams and Maunsell, 1999*; *Reynolds and Chelazzi, 2004*; *Spitzer et al., 1988*; *Winkowski and Knudsen, 2006*; *Womelsdorf et al., 2006*), we investigated whether the bat's active sonar inspection of objects in space alters the 3D sensory tuning of SC neurons. We compared the spatial receptive fields of single SC neurons when the bat produced isolated sonar vocalizations (non-SSGs) to times when it adaptively increased sonar resolution by producing SSGs (*Figure 5A* – an example trial; non-SSGs, blue circles; SSGs, red circles; *Figure 5B* –

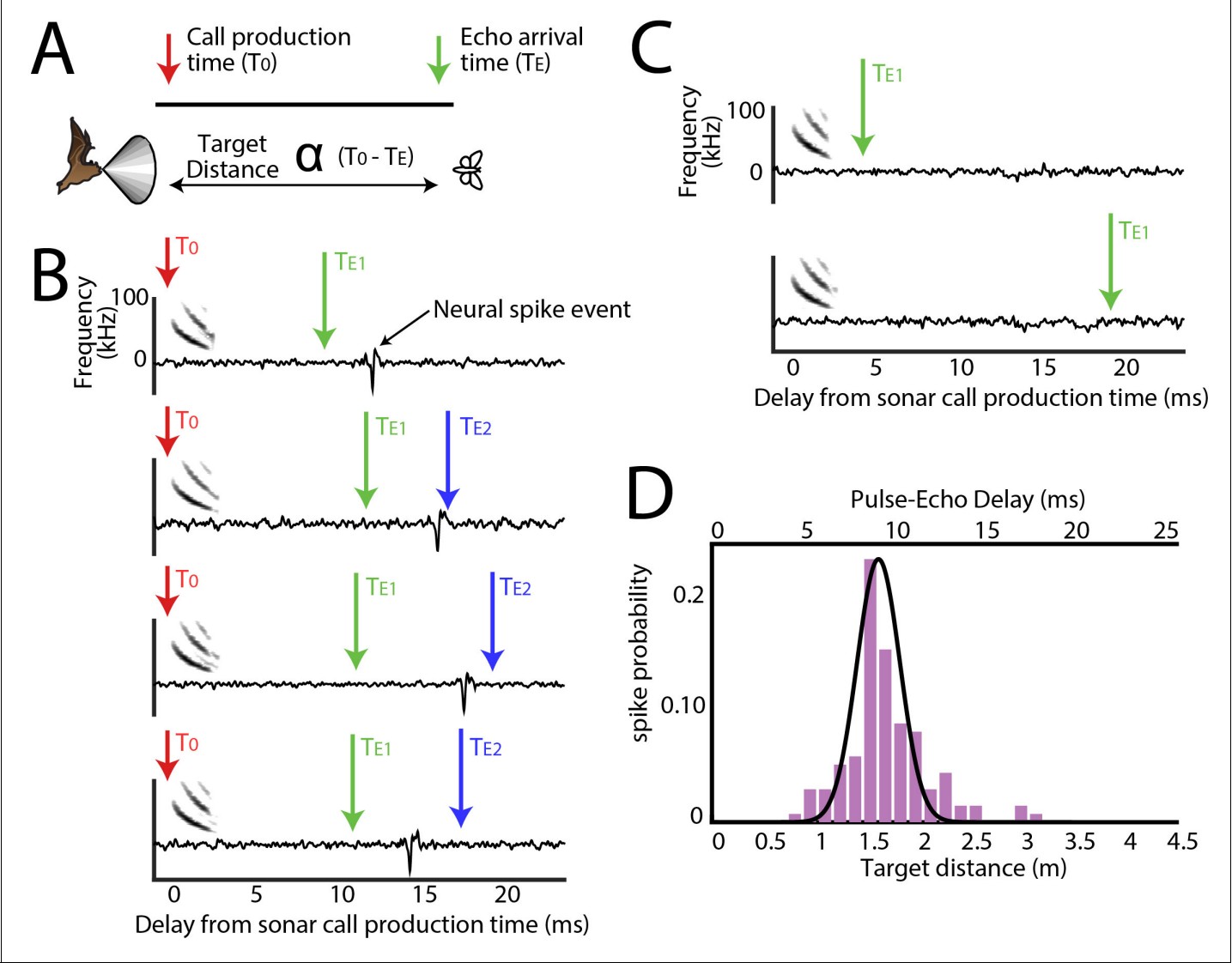

**Figure 3.** Range tuning of midbrain neurons. (**A**) A cartoon representation showing the target range estimation in a free-flying echolocating bat. The difference between the call production time (T0, red arrow) and the echo arrival time (TE, green arrow) is a function of target distance. (**B**) Sensory responses of a single neuron to echo returning at a specific delay with respect to sonar vocal onset from actual trial data. The arrival time of the first echo ($TE_1$) is indicated with a green arrow, the second echo ($TE_2$ – from a more distant object) is indicated with a blue arrow. Note that this neuron responds to the echo arriving at ~10 milliseconds. (**C**) When the echo returns at a shorter delay, the neuron does not respond; and the neuron similarly does not respond to longer pulse-echo delays. (**D**) Histogram showing target distance tuning (i.e. pulse-echo delay tuning) for the neuron in panel B and C. Note the narrow echo delay tuning curve.

DOI: https://doi.org/10.7554/eLife.29053.010

The following figure supplements are available for figure 3:

**Figure supplement 1.** Spatial coverage during the experiment.
DOI: https://doi.org/10.7554/eLife.29053.011

**Figure supplement 2.** Spatial tuning in azimuth and elevation.
DOI: https://doi.org/10.7554/eLife.29053.012

spectrograms from the data in 6A, with SSGs again highlighted in red; *Figure 5C* – a plot showing SSGs can be quantitatively identified, see Materials and methods).

We discovered that a neuron's distance tuning is sharper to echo returns from the bat's production of SSGs, as compared to responses to echoes returning from single (non-SSG) calls (*Figure 5D* shows an example neuron). *Figure 5E* shows summary data comparing the sharpness of distance

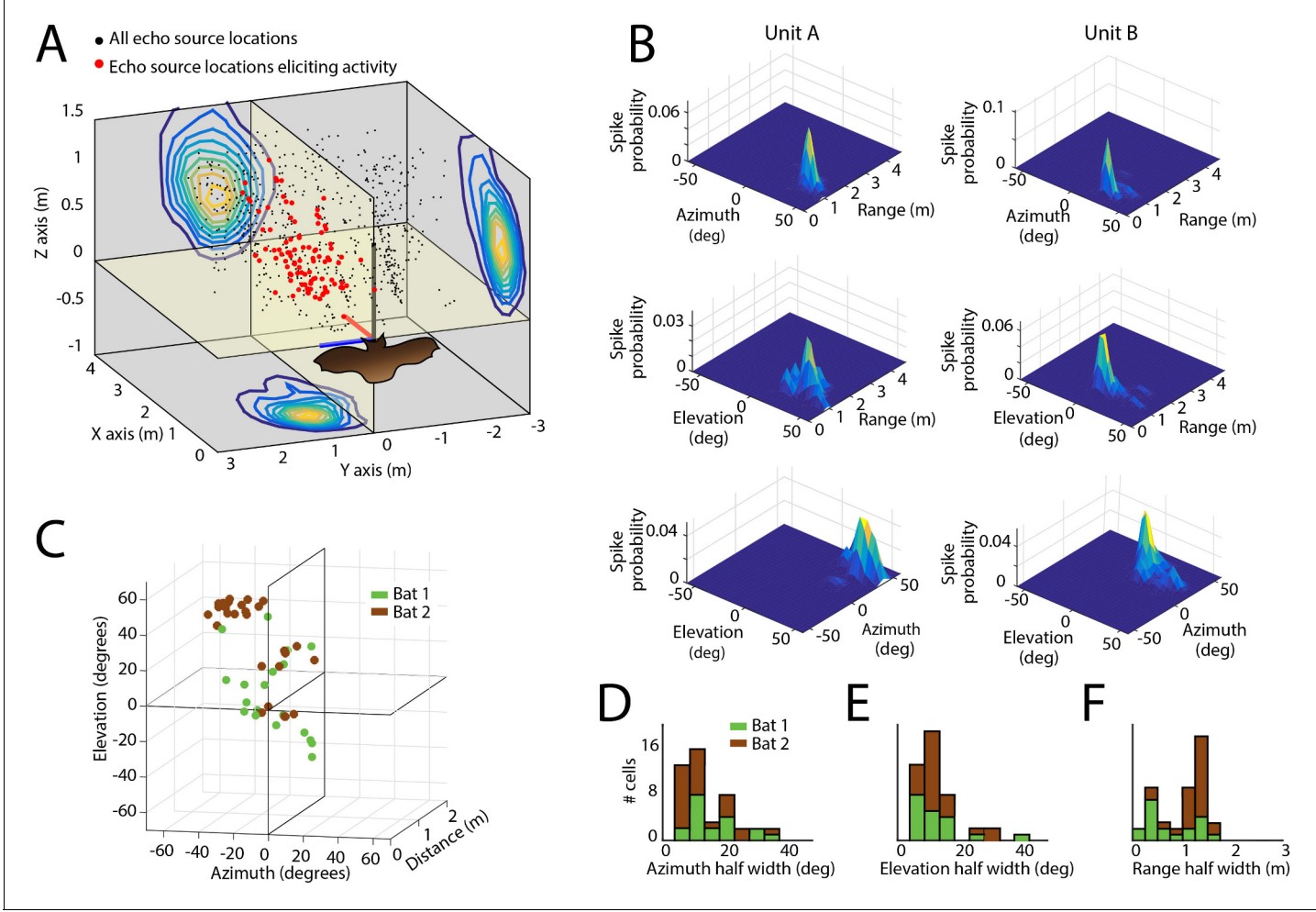

**Figure 4.** Spatial tuning of neurons recorded in the SC. (**A**) Egocentric locations of echo sources eliciting activity from a single SC neuron. Red dots indicate echo source locations eliciting spikes, black dots indicate echo source locations where a spike is not elicited. Contour plots show the XY, YZ, and ZX projections of the spatial tuning of the neuron. (**B**) 2D spatial tuning plots for two separate neurons (left column and right column). Shown are surface heat plots, where the size of the peak indicates the spike probability for a neuron for each 2D coordinate frame. (**C**) Centers of 3D spatial tuning for 46 different neurons recorded in the SC. Different bats are indicated by different colors (Bat 1 in green, Bat 2 in brown). (**D, E and F**) Left to right: azimuth, elevation, and range half width tuning properties for 46 different neurons recorded in the SC (colors as in panel C).
DOI: https://doi.org/10.7554/eLife.29053.013

The following figure supplements are available for figure 4:

**Figure supplement 1.** Stability of distance tuning across the first and last half of each recording session.
DOI: https://doi.org/10.7554/eLife.29053.014

**Figure supplement 2.** Changes in depth tuning as a function of recording depth.
DOI: https://doi.org/10.7554/eLife.29053.015

**Figure supplement 3.** Distribution of cells showing 3D, 2D and 1D spatial tuning.
DOI: https://doi.org/10.7554/eLife.29053.016

tuning to echoes returning from SSG and non-SSG calls (n = 51, neurons which met the power analysis criterion, see Materials and methods for details about power analysis; data from Bat 1 is shown in green, Bat 2 in brown). *Supplementary file 1A* – gives details of sharpness of distance tuning comparisons for SSG and non-SSG tuning, using the Brown-Forsyth test, for each of the neurons in *Figure 5E*.

We also found that a neuron's best echo delay (target distance) is often shifted to shorter delays (closer objects) when the bat is engaged in the production of SSGs, suggesting that distance tuning is dynamically remapped when the bat actively inspects objects in its environment (*Figure 5D*

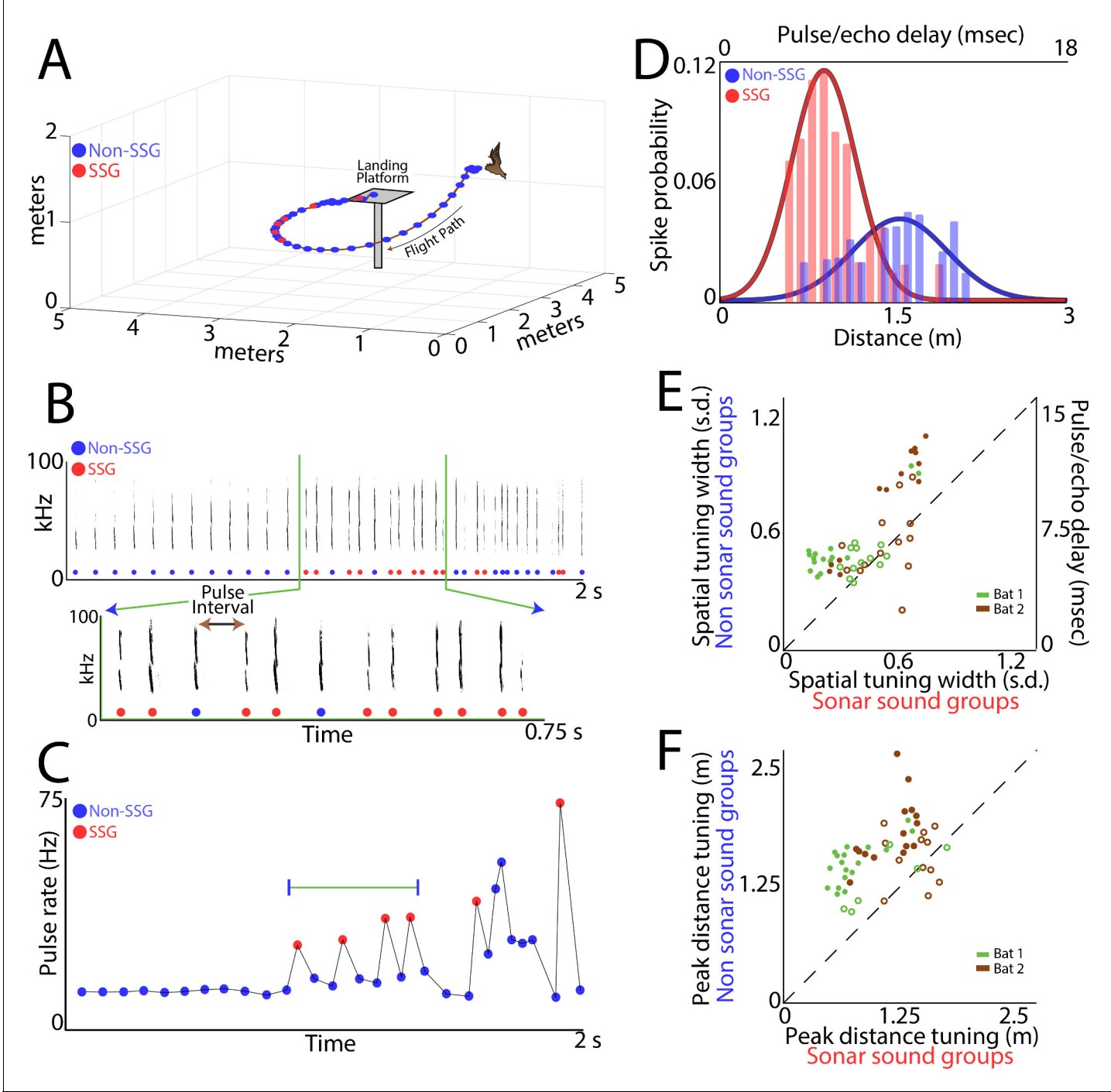

**Figure 5.** Adaptive vocal behavior drives changes in spatial tuning of SC neurons. (**A**) Three-dimensional view of one flight path (in black) through the experimental room. Individual sonar vocalizations that are not included in a sonar sound group (non-SSG) are shown as blue circles, and sonar vocalizations within a sonar sound group (SSG) shown in red. (**B**) Top, spectrogram of sonar vocalizations emitted by the bat in panel A. Bottom, expanded region of top panel to indicate SSGs and the definition of pulse interval (PI). (**C**) Change in pulse rate (1/PI) during the flight shown in panel A, and for the vocalizations shown in panel B. Note the increase in pulse rate indicative of SSG production. (**D**) Change in spatial tuning of example neuron when the bat is producing SSGs (red) as opposed to non-SSGs (blue). Note that the distance tuning decreases, as well as the width of the tuning curve, when the bat is producing SSGs. (**E**) Summary plot of change in spatial tuning width when the bat is producing SSGs (n = 53 neurons). Many single neurons show a significant sharpening (n = 26) in spatial tuning width along the distance axis when the bat is producing SSGs and listening to echoes, as compared to times when the bat is receiving echoes from non-SSG vocalizations (Bat 1 is indicated with green, Bat 2 is indicated with brown; units with significant sharpening at p<0.05 are indicated with closed circles, non-significant units indicated with open circles; Rank-Sum test). (**F**) Summary plot of change in mean peak spatial tuning when the bat is producing SSGs (n = 51 neurons). Many neurons show a significant decrease

*Figure 5 continued on next page*

*Figure 5 continued*

(n = 32) in the mean of the peak distance tuning during the times of SSG production as compared to when the bat is producing non-SSG vocalizations (Bat 1 is indicated with green, Bat 2 is indicated with brown; units with significant sharpening at p<0.05 are indicated with closed circles, non-significant units indicated with open circles; Brown-Forsythe test).

DOI: https://doi.org/10.7554/eLife.29053.017

The following figure supplement is available for figure 5:

**Figure supplement 1.** Changes in firing probability and spatial receptive fields for the first and last call of SSGs.

DOI: https://doi.org/10.7554/eLife.29053.018

example). *Figure 5F* shows summary data, comparing the mapping of distance tuning of single neurons in response to echoes from SSG and non-SSG calls (n = 53 neurons which met the power analysis criterion, see Materials and methods for details about power analysis; data from Bat 1 is shown in green, Bat 2 in brown). *Supplementary file 1B* – gives details of mean distance tuning comparisons for SSG and non-SSG echo delay responses, using the Brown-Forsyth test. For each of the neurons in *Figure 5E and F*; filled circles indicate cells with a significant sharpening (*Figure 5E*), or a significant decrease in peak distance tuning in response to echoes from SSGs (*Figure 5F*); while open circles indicate non-significant comparisons (rank-sum, p<0.05). We also examined the responses to echoes returning from the first sonar vocalization of an SSG versus the last vocalizations of an SSG. We found that there is no difference in spatial tuning profiles computed separately for the first and last echoes of SSGs, but there is a significant increase in spike probability in response to echoes from the last vocalization of an SSG (*Figure 5—figure supplement 1*).

## Gamma power increases during epochs of sonar sound group production

Similar to foveation, which is a behavioral indicator of visual attention to resolve spatial details (*Reynolds and Chelazzi, 2004*), measurements of adaptive sonar behavior have been used as a metric for the bat's acoustic gaze to closely inspect objects (*Moss and Surlykke, 2010*). Previous behavioral research shows that bats increase the production of sonar sound groups (SSGs) under conditions that demand high spatial resolution, e.g. in dense acoustic clutter and when tracking erratically moving targets (*Kothari et al., 2014*; *Moss et al., 2006*; *Petrites et al., 2009*; *Sändig et al., 2014*). SSGs are clusters of echolocation calls, often produced at stable rate (*Figure 6A*, see Materials and methods), which are hypothesized to sharpen acoustic images of objects in the environment (*Moss and Surlykke, 2010*), and are distinct from the overall increase in sonar call rate of a bat approaching a target. Previous work in other systems has shown that the gamma frequency band (40–140 Hz - *Sridharan and Knudsen, 2015*) of the LFP in the SC increases in power when an animal is attending in space (*Gregoriou et al., 2009*; *Gunduz et al., 2011*; *Sridharan and Knudsen, 2015*), and we investigated whether this conserved indicator of spatial attention also appears during SSG production. Shown in *Figure 6B* is a comparison of gamma band activity during the bat's production of SSGs over non-SSGs, demonstrating an increase around the time of SSG production. Displayed is the call triggered average (±s.e.m.) of the gamma band across recording sites, for SSG (red, n = 539) and non-SSG (blue, n = 602) production. *Figure 6C* illustrates the significant increase in gamma band power during the production of SSGs (red) as compared to non-SSGs (blue) on a neuron-by-neuron basis (n = 26), and this finding was consistent across recording depths (*Figure 6—figure supplement 1*). Only sites in which neural recordings were unaffected by motion artifact were included in this analysis (*Figure 6—figure supplement 2*, Also see Materials and methods). In agreement with past work in other systems and brain areas (*Gregoriou et al., 2009*; *Gunduz et al., 2011*; *Sridharan and Knudsen, 2015*), there was a significant increase in gamma power when the bat produced SSGs, providing further evidence that SSGs indicate times of sonar inspection and spatial attention (*Figure 6C*, p<0.005, Wilcoxon sign-rank test).

Additionally, we analyzed the timing of gamma power increase with respect to echo-evoked neural activity. Because sensing through echolocation temporally separates vocal production time from echo arrival time, we can accurately measure the amplitude of gamma activity with respect to motor production and/or sound reception. The data show that the increase in gamma power occurred specifically around the time of the echo-evoked spike events in SC sensory neurons (*Figure 6D* – SSGs

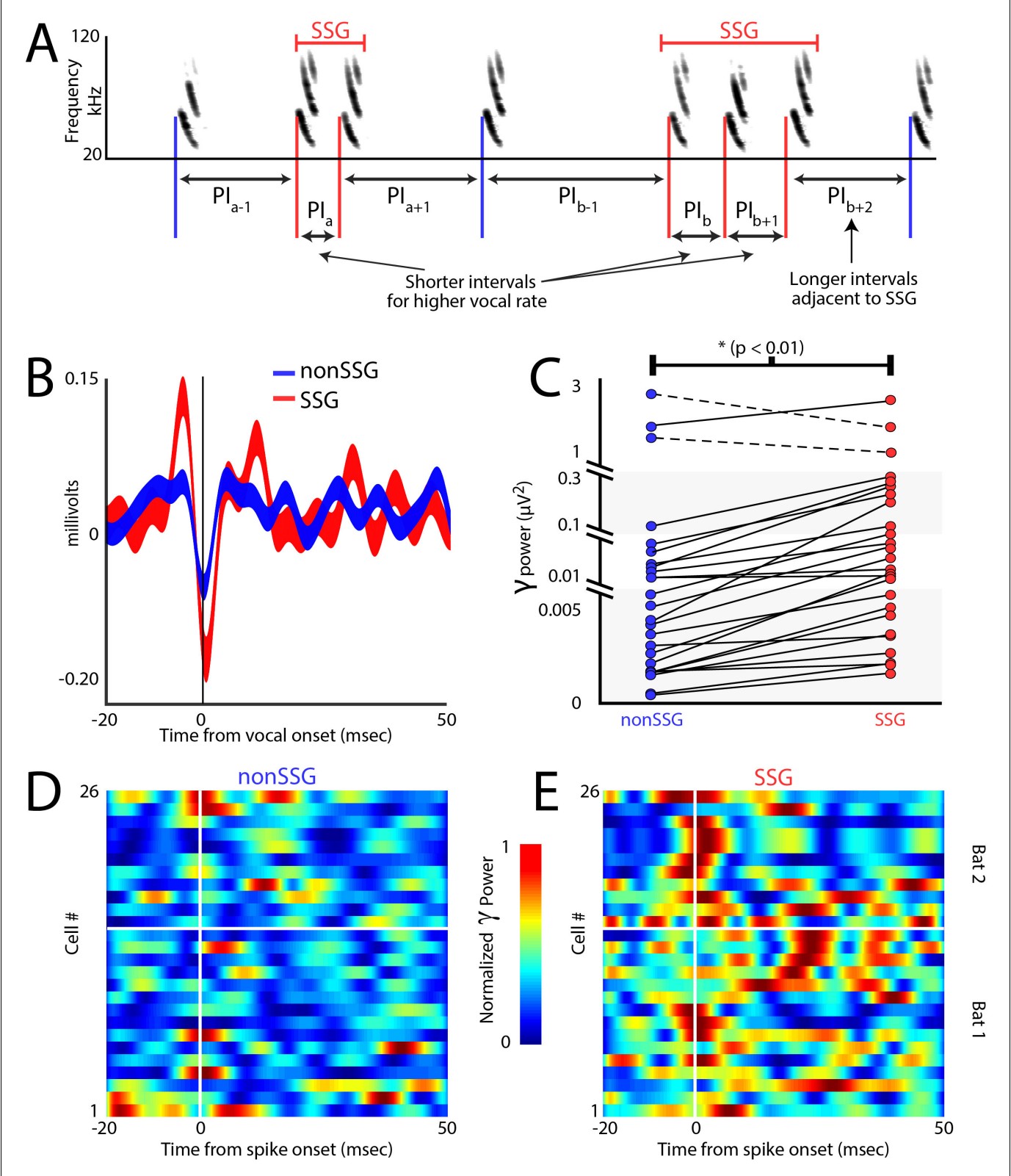

**Figure 6.** Increases in gamma power correlate with sonar-guided spatial attention. (A) Schematic of sonar sound group (SSG) determination. SSG's are identified by brief epochs of higher vocal rate (i.e. shorter interval in red) surrounded by vocalizations at a lower rate (i.e. longer interval in blue). (B) Average gamma waveform at the onset of single sonar vocalizations, or non-SSG's (blue, n = 26), compared to the average gamma waveform at the onset of vocalizations contained within an SSG (red, n = 26). Plotted is the mean ±s.e.m. (C) Pair-wise comparison of power in the gamma band during

*Figure 6 continued on next page*

*Figure 6 continued*
the production of non-SSG vocalizations (blue) and SSG vocalization (red). There is a significant increase in gamma power during SSG production across neurons (n = 26, Wilcoxon sign-rank rest, p<0.01). (D) Normalized increase in gamma power at the time of auditory spike onset for each neuron during the production of non-SSG vocalizations. (E) Normalized increase in gamma power at the time of auditory spike onset for each neuron during the production of SSG vocalizations. Note the higher gamma power during SSG production, and the temporal coincidence of the increase in gamma with spike time (vertical white line indicates spike time, horizontal white line separates data from Bat 1, below and Bat 2, above.
DOI: https://doi.org/10.7554/eLife.29053.019
The following figure supplements are available for figure 6:

**Figure supplement 1.** Changes in gamma band power ratio for SSG and non-SSGs with recording depth.
DOI: https://doi.org/10.7554/eLife.29053.020
**Figure supplement 2.** Wing beat motion artifact in LFP.
DOI: https://doi.org/10.7554/eLife.29053.021

and 6E – non-SSGs, vertical white line indicates onset of sensory evoked spikes, horizontal white line separates data from Bat 1, below, and Bat 2, above), and that the increase in gamma band power is temporally precise, with the peak in gamma power occurring within 10 milliseconds of spike time.

## Discussion

Spatially-guided behaviors, such as obstacle avoidance, target tracking and reaching, all depend on dynamic egocentric sensory representations of the 3D positions of objects in the environment. An animal must not only compute the direction and distance to targets and obstacles, but also update this information as it moves through space. How does the nervous system of a freely moving animal encode 3D information about the location of objects in the physical world? And does active inspection of objects in the environment shape 3D sensory tuning? Our neural recordings from the midbrain of a freely moving animal engaged in natural, spatially-guided behaviors offer answers to these fundamental questions in systems neuroscience.

Here we present the first characterization of 3D sensory responses of single neurons in the midbrain SC of an animal actively interacting with its physical environment. We also show that echo-evoked spatial tuning of SC neurons sharpens along the range axis and shifts to closer distances when the bat inspects objects in its acoustic scene, as indexed by the production of sonar sound groups (SSGs) (*Falk et al., 2014*; *Kothari et al., 2014*; *Moss et al., 2006*; *Petrites et al., 2009*; *Sändig et al., 2014*). It has been hypothesized that the bat produces SSGs to enhance spatial resolution, in a manner similar to foveal fixation, which increases visual resolution (*Moss and Surlykke, 2010*; *Surlykke et al., 2016*). Our data provide the first empirical evidence of sharpened 3D spatial resolution of single neurons in the bat's auditory system with natural and dynamic adaptations in the animal's active orienting behaviors.

### Role of the SC in 3D spatial orientation

The superior colliculus (SC), a midbrain sensorimotor structure, is implicated in species-specific sensory-guided orienting behaviors, target selection and 2D spatial attention (*Duan et al., 2015*; *Knudsen, 2011*; *Krauzlis et al., 2013*; *Lovejoy and Krauzlis, 2010*; *McPeek and Keller, 2004*; *Mysore and Knudsen, 2011*; *Mysore et al., 2011*; *Zénon and Krauzlis, 2012*). Past research has led to conflicting views as to whether the SC plays a role in orienting in 3D space (*Chaturvedi and Gisbergen, 1998*; *Chaturvedi and van Gisbergen, 1999*; *Chaturvedi and Van Gisbergen, 2000*; *Hepp et al., 1993*; *Van Horn et al., 2013*; *Leigh and Zee, 1983*; *Walton and Mays, 2003*), but limited evidence from sensory mapping in primates shows response selectivity to binocular disparity (*Berman et al., 1975*; *Dias et al., 1991*), and vergence eye movements (*Chaturvedi and Gisbergen, 1998*; *Chaturvedi and van Gisbergen, 1999*; *Chaturvedi and Van Gisbergen, 2000*; *Van Horn et al., 2013*), indicating a role of the SC in 3D visuomotor integration. Here, we present the first direct evidence of *3D egocentric sensory* responses to physical stimuli in the midbrain of an animal freely moving through its environment. Our results therefore provide a critical bridge to understanding the brain's dynamic representation of the 3D physical world.

## Behavioral and neural correlates of spatial attention

Psychophysical studies have reported that human and non-human primates show increased visual detection and discrimination performance when stimuli are presented at attended locations (*Bichot et al., 2005*; *Carrasco, 2011*; *Posner, 1980*; *Wurtz and Mohler, 1976*; *Yeshurun and Carrasco, 1999*). Neural recording experiments have corroborated these results by showing that spatial attention modulates firing rates of cortical neurons representing attended locations (*McAdams and Maunsell, 1999*; *Reynolds and Chelazzi, 2004*; *Reynolds et al., 1999*; *Spitzer et al., 1988*; *Womelsdorf et al., 2006*). Other studies report an increase in the gain of tuning curves at an attended location or a selected stimulus feature, while a decrease in neural response occurs for unattended locations or features (*McAdams and Maunsell, 1999*; *Treue and Martínez Trujillo, 1999*; *Verghese, 2001*).

The midbrain SC has been specifically implicated in an attention network through past studies of SC inactivation that produced behavioral deficits (*Lovejoy and Krauzlis, 2017*; *McPeek and Keller, 2004*), but none of these studies measured the spatial selectivity of single SC neurons under conditions in which animals freely inspected objects in the physical environment. Evidence for sharpening of tuning curves and/or remapping spatial receptive fields with attention has been limited to a few studies showing shifts in 2D cortical tuning to artificial visual stimuli in restrained animals (*Spitzer et al., 1988*; *Womelsdorf et al., 2006*). And in studies of the auditory system, behavioral discrimination of acoustic stimuli has been shown to influence the response profiles of cortical neurons in restrained ferrets (*Fritz et al., 2003*, *2007*). Here we report for the first time dynamic shifts in 3D sensory tuning with sonar-guided attention in animals engaged in natural orienting behaviors.

Our study not only revealed changes in single neuron 3D spatial selectivity with dynamic sonar inspection of objects in the physical scene, but also a corresponding increase in the gamma band of the local field potential (LFP). Past work in humans, non-human primates, other mammals, and birds have reported stimulus driven gamma band modulation when stimuli are presented at attended locations (*Fries et al., 2001*; *Goddard et al., 2012a*; *Gregoriou et al., 2009*; *Sridharan and Knudsen, 2015*; *Sridharan et al., 2011*). Moreover, changes in the gamma band of the LFP have been shown to occur for stimulus selection and discrimination mediated by touch, vision, and hearing, suggesting that gamma oscillations may reflect multi-modal network activity related to attention (*Bauer et al., 2006*; *Canolty et al., 2006*; *Gruber et al., 1999*; *Senkowski et al., 2005*). Our findings that gamma power increases during epochs of SSG production and echo reception support the hypothesis that the bat's adaptive sonar behaviors serve as indicators of spatial attention (*Moss and Surlykke, 2010*).

## 3D allocentric versus 3D egocentric representations in the brain

It is important to emphasize the distinction between our report here on 3D *egocentric sensory responses* in the midbrain SC of the insectivorous echolocating big brown bat, and 3D *allocentric memory-based representation* of space in the hippocampus of the echolocating Egyptian fruit bat (*Yartsev and Ulanovsky, 2013*). These two distinct frames of reference are used for different suites of natural behaviors. *Egocentric sensory representation* of space contributes to overt and covert orienting to salient stimuli (*Knudsen, 2011*; *Krauzlis et al., 2013*; *Mysore and Knudsen, 2011*) and has not previously been described in free-flying bats. By contrast, 3D allocentric (*Geva-Sagiv et al., 2015*; *Yartsev and Ulanovsky, 2013*) and vectorial representations (*Sarel et al., 2017*) in the bat hippocampus support spatial memory and navigation. Further, published studies on the Egyptian fruit bat hippocampus have not considered the acoustic sensory space of this species that uses tongue clicks to echolocate (*Yovel et al., 2010*), nor potential modulation of hippocampal activity by sonar signal production. In other words, past work on the Egyptian fruit bat hippocampus shows 3D spatial memory representation; whereas, our study of the big brown bat SC reveals important new discoveries of state-dependent midbrain sensory representation of 3D object location.

## Depth tuning of single neurons in the bat auditory system

Finally, and importantly, our results fill a long-standing gap in the literature on the neural representation of target distance in the bat auditory system, which has almost exclusively been studied in passively listening animals (*Dear and Suga, 1995*; *Feng et al., 1978*; *O'Neill and Suga, 1979*; *Valentine and Moss, 1997*), but see *Kawasaki et al., 1988*. Echolocating bats estimate target

distance from the time delay between sonar call emission and echo reception, and show behavioral range discrimination performance of less than 1 cm, which corresponds to an echo delay difference of about 60 μsec (*Moss and Schnitzler, 1995*; *Simmons, 1973*). The bat's sonar signal production is therefore integral to target ranging, and yet, for over nearly four decades of research, scientists have simulated the dimension of target distance in neural recording experiments in restrained bats by presenting pairs of synthetic sound stimuli (P/E pairs – pulse/echo pairs), one mimicking the echo-location call, and a second, delayed and attenuated signal, mimicking the echo. Here, we report the first delay-tuned neural responses to echoes from physical objects in the auditory system of free-flying bats, thus providing a critical test of a long-standing hypothesis that neurons in actively echolo-cating bats respond selectively to echoes from objects in 3D space.

*Beetz et al. (2016a)* report that distance tuning of neurons in the auditory cortex of passively lis-tening, anesthetized bats (*Carollia perspicillata*) is more precise when neurons are stimulated with natural sonar sequences, such as those produced by echolocating bats in the research reported here. Another study of auditory cortical responses in anesthetized bats (*Phyllostomus discolor*) reports that delay-tuned neurons shift their receptive fields under stimulus conditions that simulate echo flow. (*Bartenstein et al., 2014*). In a related study, *Beetz et al., 2016b* show a higher probabil-ity of neural firing in cortical neurons of the bat species *Carollia perspicillata* to the first echo in a sequence, which leads them to hypothesize that global cortical inhibition contributes to the repre-sentation of the closest object, without active attention. It is possible that global cortical inhibition is an intrinsic feature, which enables an animal to represent the most salient (in the above case, closest) stimulus. Our data also show that sensory neurons respond primarily to the first echo arriving in a neuron's receptive field, as compared to later echoes, and may depend on a similar mechanism. A mechanism of global inhibition for selective attention has also been demonstrated in the barn owl optic tectum (*Mysore et al., 2010*). Additionally, our data demonstrate a higher probability of audi-tory responses in the midbrain SC to echoes returning from the last echo of a SSG, a finding, which can only be demonstrated in a behaving echolocating bat, as it involves feedback between sensing and action. And while studies of auditory cortical processing in anesthetized, passively listening ani-mals can shed light on sensory processing mechanisms, ultimately this information must be relayed to sensorimotor structures, such as the midbrain superior colliculus, which serve to orchestrate appropriate motor commands for spatial navigation and goal-directed orientation.

Our study reveals the novel finding that auditory neurons in awake and behaving echolocating bats show shifts and sharpening of spatial receptive fields with echolocation call dynamics. Crucially, because bats in our study were engaged in a natural spatial navigation task, we could directly inves-tigate the effects of sonar-guided attention on the 3D spatial tuning of single auditory neurons. Our results demonstrate the dynamic nature of 3D spatial selectivity of single neurons in the SC of echolocating bats and show that active behav-ioral inspection of objects not only remaps range response areas, but also sharpens depth tuning. Furthermore, our data reveal echo-delay tuning of single SC neurons in response to echoes from actively echolocating bats is sharper than previ-ously reported from recordings in passively lis-tening bats (*Dear and Suga, 1995*; *Menne et al., 1989*; *Moss and Schnitzler, 1989*; *Simmons et al., 1979*; *Simmons et al., 1990*; *Valentine and Moss, 1997*) and bear rele-vance to a long-standing controversy on the neu-ral basis of fine echo ranging acuity of bats (*Menne et al., 1989*; *Moss and Schnitzler, 1989*; *Simmons, 1979*; *Simmons et al., 1990*).

In summary, our study generated new discov-eries in the field of systems neuroscience by inte-grating chronic neural recordings, multimedia tracking of dynamic animal behaviors in the 3D

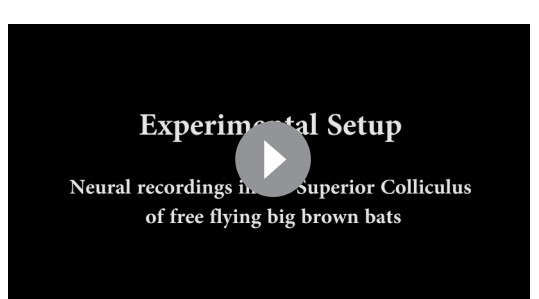

**Video 1.** Experimental setup for validating the echo model. This is a two-part movie. The first part shows the layout of the microphone array, which is used to capture the sonar vocalizations of the bat as it flies and navigates around objects in its path. For simplicity only two objects are shown here. The second part of the movie shows the use of the 14-channel echo microphone array, which captures the returning echoes as the bats flies in the forward direction. Note that the echo microphone array is placed behind the bat on the wall opposite to its flight direction.
DOI: https://doi.org/10.7554/eLife.29053.022

physical environment, and acoustic modeling. We report here the first empirical demonstration that neurons in a freely moving animal encode the 3D egocentric location of objects in the real world and dynamically shift spatial selectivity with sonar-guided attention. Specifically, we show that single neurons in the actively echolocating, free-flying bat respond selectively to the location of objects over a restricted distance (echo delay), azimuth and elevation. Importantly, we discovered that the sensory response profiles of SC neurons become sharper along the range axis and shift to shorter distances (echo delays) when the bat actively inspects physical objects in its environment, as indicated by temporal adjustments in its echolocation behavior. Our discovery of dynamic 3D sensory representations in freely behaving animals call for comparative studies in other species, which can collectively contribute to a more complete understanding of nervous system function in the context of natural behaviors.

## Materials and methods

### Bats

Two adult big brown bats, *Eptesicus fuscus*, served as subjects in this study. Bats were wild caught in the state of Maryland under a permit issued by the Department of Natural Resources and housed in an animal vivarium at the University of Maryland or Johns Hopkins University. Both the University of Maryland's, and Johns Hopkins University's Institutional Animal Care and Use Committee approved all of the procedures utilized for the current study.

### Experimental design

The two big brown bats were tested in related tasks, carried out in a 6 × 6 × 2.5 m room, illuminated with IR and equipped with 16 high-speed cameras and an ultrasound microphone array (*Figure 1*, see below). The first bat navigated around objects in a large flight room and landed on a platform. In order to ease the task for the second bat, it simply flew around the room, navigated around objects, and landed on any wall. Both bats were fed mealworms at the end of each trial to keep them active, but they were not rewarded for flight. The flight room was illuminated with infrared lighting (~850 nm) to preclude the bat's use of vision, ERG data show that *Eptesicus* does not see wavelengths longer than 600 nanometers (*Hope and Bhatnagar, 1979*). The room was also equipped with high-speed cameras and an ultrasound microphone array to track the bat's flight path and record the bat's echolocation behavior. Bats navigated around obstacles in the room (explained in detail below), and were released at different locations in the room for each trial (eight positions for Bat 1, five different positions for Bat2), which required them to use sonar echoes to steer around obstacles rather than a consistent or memorized flight path around objects in the room (see *Figure 3—figure supplement 1A*). As such, the bats determined the duration and flight path of each trial. The obstacles were four plastic cylinders (hard plastic as to be acoustically reflective), approximately 13 cm in diameter and 30 cm in length.

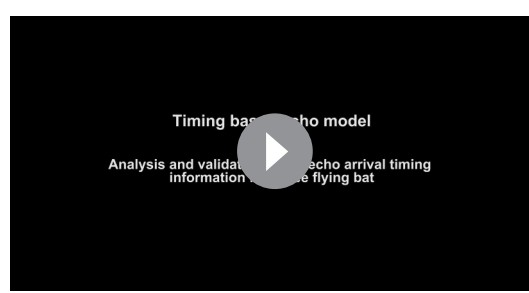

**Video 2.** Validation of echo model using time-difference-of-arrival (TDOA) algorithms. This is a two-part movie. The first part consists of 3 panels. The top panel shows an example trajectory as the bat navigates across objects (white and green). The red line is the reconstructed trajectory and green circles along the trajectory are positions where the bat vocalized. The center and bottom panels are time series when the bat vocalizes and when echoes arrive at the bat's ears, respectively. The echo arrival times have been computed using the echo model. The second part of the movie demonstrates the localizations of echo sources using TOAD algorithms. This movie has four panels. The top left panel shows the spectrogram representation of the recording of the bat's vocalizations. The left center and bottom panels show spectrograms of 2 channels of the echo microphone array. The right panel shows the reconstructed flight trajectory of the bat. Echoes received on four or more channels of the echo microphone array, are then used to localize the 3D spatial location of the echo sources. These are then compared with the computations of the echo model and lines are drawn from the microphones to the echo source if the locations are validated.
DOI: https://doi.org/10.7554/eLife.29053.023

Once the bat flew freely throughout the room and in the case of Bat 1, learned to land on a platform, a surgery was performed to implant in the midbrain superior colliculus (SC) a 16-channel chronic recording silicon probe (Neuronexus) mounted on a custom microdrive. The bats' weights were between 18 and 21 grams, and the weight of the implant, microdrive and transmitter device was 3.8 grams. The bat was given several days to rest and acclimate to the implanted device, after which they were able to fly and navigate around objects in the flight room. Data collection began after the animal was able to perform ~30 flight trials per session, which took place twice a day (morning and afternoon) in the experimental test room. During experimental sessions, there was no conditional reward; instead the bats were fed mealworms at the end of every trial, that is, when they landed. Bat 1 flew for 12 sessions, and Bat 2 flew for 15 sessions. For each recording session, the positions of the four flight obstacles were varied. Further, across trials the bat was released from different locations in the room. The obstacle configurations and flight start locations were varied to ensure that the bat's flight trajectories covered the entire room, and the stimulus space sampled by the bat changed from trial to trial. This approach prevented the bats from relying on spatial memory and/or stereotyped flight paths. *Figure 3—figure supplement 1A* shows the bat's flight trajectories in a single session and illustrates room coverage. Coverage was restricted in elevation, due to the height of the flight room, with a floor to ceiling dimension of approximately 250 cm. Although the landing behavior of the bats differed slightly (i.e. landing on a platform vs. a wall), neural analysis was focused on the times when the animals were in flight and the data from the two bats are comparable. Additionally, both bats performed natural echolocation and flight behaviors as neural recordings were taken.

## Video recording

The flight trajectory of the bat was reconstructed using a motion tracking system with 16 high-speed cameras (Vicon). The motion tracking system was calibrated with a moving wand-based calibration method (*Theriault et al., 2014*), resulting in sub-millimeter accuracy and 3D spatial location information of the bat at a frame rate of 300 Hz. Once the motion tracking system is calibrated, it tracks the bat in a 3D coordinate frame of reference, which we refer to as 'world coordinates.' Affixed on the dorsal side of the transmitter board were three IR reflective markers (3 mm round) that were then tracked with the high-speed motion tracking system (Vicon). By tracking the 3D position of these three markers, we were able to determine the 3D position and head aim of the bat during the experiment. Around the perimeter of the room, at a distance from the walls of about 0.5 meters, the motion capture cameras did not provide adequate coverage, and data from the bat at these locations was not used for analysis.

## Audio recordings

In addition to recording the position of the bat, we also recorded the sonar calls of the bat using an array of ultrasonic microphones (Pettersson Elektronik, Ultrasound Advice, see *Figure 1A*). The microphone recordings were hardware bandpass filtered between 10 KHz and 100 KHz (Alligator Technologies and Stanford Research Systems) and were digitized using data acquisition systems (National Instruments + custom built hardware).

## Synchronization of systems

All three hardware systems (i.e. neural recording, video-based 3D positioning, and microphone array) were synchronized using the rising edge of a square pulse generated using a custom circuit. The square pulse was manually triggered at the end of each trial (i.e. at the end of each individual flight) when the bat landed on the platform/wall. At the generation of the TTL pulse, each system (video and audio) saved 8 s of pre-buffered data into the hard disk of the local computer.

## Analysis of flight behavior

To ensure that the bats were not using spatial memory to guide their flight, we randomly released the bats from different spatial locations in the flight room. The average number of flights per session were 22 for Bat 1 and 27 for Bat 2. Further, we used eight positions (a-h) for releasing Bat 1 and 6 positions (a-f) for releasing Bat 2. To evaluate stereotypy in the bats' flight paths, we used methods previously developed by *Barchi et al., 2013*. Occupancy histograms were created by collapsing the

3D trajectory data to 2D plan projection (*x,y* and *x,z*). The number of points across a set of flight paths that fell inside 10 cm$^2$ bins were counted. These points were converted to probabilities by dividing each bin count by the total number of points across each set of flights. After normalization, the occupancy histograms of trials could be compared within each session. The next step was to compute the autocorrelation of each trial and cross-correlation of each trial with every other trial. The maximum value of each 2D cross-correlation was divided by the maximum value of the autocorrelation. This ratio is shown as a matrix for a representative session for both bats in *Figure 1—figure supplement 1*. The value of each square along the diagonal is one (yellow on the color bar) as it represents the autocorrelation of each flight trajectory. Cooler colors indicate minimum correlation between flight trajectories and warmer colors indicate stereotypy between trajectories.

## Surgical procedures, neural recordings and spike sorting

Once the bats were trained on the task, a surgery was performed to implant a 16-channel silicon probe (Neuronexus). The probe consisted of four shanks spaced 100 µm micrometers apart, with four recording sites also spaced 100 µm apart on each shank, resulting in a 300 × 300 square µm grid of recording sites. The silicon probe was connected by a ribbon cable to an electrical connector (Omnetics), and this assembly was then mounted on a custom-made, manual microdrive so that it could be moved through the dorsal/ventral axis (i.e. across layers) of the superior colliculus during the experiment. The silicon probe and microdrive assembly was then mounted on the head of the bat over a craniotomy performed above the superior colliculus (SC). The SC sits on the dorsal surface of the brain of the big brown bat (*Valentine and Moss, 1997*; *Valentine et al., 2002*), allowing for skull surface landmarks to be used in determining the implant location. Once the recording implant was positioned, a cap was made with cyanoacrylate (Loctite 4013) to protect and secure the implant to the skull surface. The bat was allowed several days to recover, and then we started running the neural recording experiment.

In order to study neural activity in the superior colliculus during a real-world navigation task, a wireless neural-telemetry system (Triangle BioSystems International) was used in conjunction with a multi-channel neural acquisition platform (Plexon). This allowed for chronic neural recordings to be collected from the superior colliculus (SC) while the echolocating bat was navigating around obstacles in flight. During the experiment, a wireless RF telemetry board (Triangle BioSystems International) was connected to the plug of the silicon probe mounted on top of the bat's head. Bat 1 flew for 12 sessions while recordings were made in the SC, and Bat 2 flew for 15 sessions. Each session typically lasted 30–45 min, and the microdrive was advanced at the end of each session to collect activity from a new set of neurons in the following recording session.

Neural data were sorted offline after filtering between 800 and 6000 Hz using a 2$^{nd}$ order elliptic filter. Filtered neural traces were then sorted using a wavelet based algorithm and clustering technique (*Quiroga et al., 2004*). Furthermore, we determined the L$_{ratio}$ and isolation distance for each wavelet-based cluster in order to provide a traditional measure of the efficacy of our clustering technique. In previous reports, an L$_{ratio}$ less than 0.07, and an isolation distance more than 15, were used as thresholds for significantly separated spike-waveform clusters (*Saleem et al., 2013*; *Schmitzer-Torbert et al., 2005*). For our wavelet-based clustering technique, all L$_{ratio}$'s were less than 0.05, and isolation distances were greater than 15 (*Figure 1—figure supplement 3*), providing a secondary quantitative metric of the significant separation of our single unit clustering.

This algorithm also separated movement artifact out of the raw neural traces. If any spike events occurred simultaneously with movement artifact, however, they were not recoverable. Movement artifact rarely occurred across all channels during flight and was mostly confined to times when the bat was landing. We only used data from the bats in flight for analysis. Of all sorted single units (n = 182), 67 units (sensory neurons) were selected for analysis, as described below. The isolated single units were stable throughout the session (see *Figure 1—figure supplement 2*).

## Analysis of audio recordings

Audio recordings were analyzed using custom Matlab software to extract the relevant sound features, that is, pulse timing, duration, and interval. Combining the pulse timing (time when sound reached a stationary microph one) with the 3D flight trajectory data allowed compensating for the

sound-propagation delays and calculating the actual call production times at the source (i.e. the veridical time when the bat produced the sonar sound).

## Identification of sonar sound groups

Sonar sound groups (SSGs) are defined as clusters of two or more vocalizations which occur at a near constant PI (within 5% error with respect to the mean PI of the sound group), and are flanked by calls with a larger PI at both ends (at least 1.2 times larger) (*Kothari et al., 2014*; *Moss and Surlykke, 2001*; *Moss et al., 2006*). SSGs of two vocalizations are also produced by the bat, and our criteria for these SSGs is that surrounding PI's must be at least 1.2 times larger than the PI between the two vocalizations contained within the SSG. Here, we use the same definitions and thresholds as used in prior work (see *Figure 6A* for a visual explanation). As we use pulse rate in the main text, it is important to note that Pulse Interval = 1/Pulse Rate.

## Echo model

The 'echo model' is an acoustic model, which takes into account the instantaneous 3D position of the bat, 3D positions of the objects, the bat's head direction vector, time of production of the sonar sound as well as the physical parameters of sound in air, in order to compute the direction and time of arrival of echoes at the bat's ears. For this model, each time the bat vocalized, we computed the arrival time and direction of returning echoes.

*Figure 2—figure supplement 1A* shows an outline of a bat with the neural telemetry headstage (TBSI). The headstage is shown as a grey box with a 16-channel Omnetics connector (male and female) at the bottom. Three reflective markers (4 mm diameter), P, Q and R (black), which are tracked by the infrared motion tracking cameras (Vicon) are also shown. A top view (cartoon) of the bat and telemetry headstage, with markers is shown in *Figure 2—figure supplement 1B*.

### Reconstruction of 3D flight trajectory, head aim and egocentric axes

The bat's flight trajectory was reconstructed by computing the centroid (geometric center) of the three markers on the head stage. In case of missing points, only the points visible to the motion tracking system were used. The three points (P, Q, R) on the head stage were arranged as a triangle, with two of the points (Q and R) at the trailing edge of the headstage (*Figure 2—figure supplement 1A and B*), and marker P at the front of the headstage. The 3D head aim of the bat was computed by first calculating the midpoint (P') of $\overline{QR}$ and then constructing $\overrightarrow{PP'}$ along the mid line of the head (*Figure 2—figure supplement 1B*, head aim vector is shown as a dashed red arrow).

$$\hat{p}_x = \frac{\overrightarrow{PP'}}{\left|\overrightarrow{PP'}\right|} \ (head \ aim \ unit \ vector) \tag{1}$$

The z-direction of the egocentric axes was computed as the cross product of $\overrightarrow{PQ}$ and $\overrightarrow{PR}$.

$$\hat{p}_z = \frac{\overrightarrow{PQ} \ X \ \overrightarrow{PR}}{\left|\overrightarrow{PQ}\right| \ X \ \left|\overrightarrow{PR}\right|} \tag{2}$$

Further, the y-direction of the egocentric axes was computed as the cross product of $p_x$ and $p_z$.

$$\hat{p}_y = \hat{p}_z \ X \ \hat{p}_x \tag{3}$$

Where X denotes cross product between vectors.

We refer to the above instantaneous egocentric coordinate system ($p_x$, $p_y$, $p_z$1) as the 'local' coordinate system and the coordinate system from the frame of reference of the motion capture cameras as the 'world' coordinate system ($P_X$, $P_Y$, $P_Z$1). An example of a reconstructed flight trajectory is shown in *Figure 2C*. This trajectory is in the 'world' coordinates shown as the X, Y, Z axes (red, green and blue colors respectively) at the left corner of *Figure 2C*. The bat's head aim during vocalizations (solid yellow circles on the flight trajectory) is indicated by black lines. *Figure 2C* also shows two example points, P($x_1$, $y_1$, $z_1$) and Q($x_2$, $y_2$, $z_2$), in the bat's flight trajectory when the bat produces

sonar calls. [p$_x$, p$_y$, p$_z$] and [q$_x$, q$_y$, q$_z$] (red, green, blue respectively) are the axes which form the 'local' instantaneous egocentric coordinate system (computed as per *Equations 1, 2 and 3*) with respect to the bat's current position in space and head aim.

To compute the instantaneous microphone, object and room boundary coordinates from the 'world' coordinate system to the 'local' instantaneous egocentric coordinate system, translation and transformation of points are performed using quaternion rotations (*Altmann, 2005*).

For example, if A(X$_a$, Y$_a$, Z$_a$) are the coordinates of an object in the global coordinate system $(P_X, P_Y, P_Z)$. Then the new coordinates A(x$_a$,y$_a$,z$_a$) of the same object with respect to the instantaneous egocentric coordinate system $(p_x, p_y, p_z)$ are computed as below (4).

$$A(x_a, y_a, z_a) = ROT(P_X, P_Y, P_Z)(p_x, p_y, p_z)(X_a, Y_a, Z_a) \tag{4}$$

## Steps to compute direction and time of arrival of echoes at the bats ears

Once the Euclidian object coordinates are transformed into the instantaneous Euclidian coordinate system $A(x_a, y_a, z_a)$, unit vectors of object directions are computed (5) and the direction angles of echo source locations can be computed by transforming from the Euclidian coordinates to spherical coordinate $A(\theta, \varphi, R)$ (azimuth, elevation, range) as given in (6).

$$\hat{a} = \frac{A(x_a, \overrightarrow{y_a}, z_a)}{\left| A(x_a, \overrightarrow{y_a}, z_a) \right|} \ (unit \ vector) \tag{5}$$

The range of the object is simply the distance between the bat's instantaneous location and the object.

$$\boldsymbol{\theta} = \sin^{-1}(\hat{a}.\hat{p}_x) \ and \ \boldsymbol{\varphi} = \sin^{-1}(\hat{a}.\hat{p}_y), \ \boldsymbol{Range(R)} = \left| A(x_a, \overrightarrow{y_a}, z_a) \right| \tag{6}$$

Time of arrival of echoes at the bat's ear is computed as given in (7).

$$T_{arr} = 2 * \frac{R}{c_{air}}, \ where \ c_{air} \ is \ the \ speed \ of \ sound \ in \ air \tag{7}$$

*Figure 2D* shows how the instantaneous solid angle of the bat's head aim vector to each object changes as the bat flies through the room. The data here refers to the flight trajectory shown in *Figure 2C*. *Figure 2E* shows the echo arrival times at the bat's ears as computed by the echo model. *Figure 2F and G* show the room, objects and microphones from the bat's egocentric point of 'view' as computed using the echo model. These figures correspond to the highlighted points, P and Q, in *Figure 2C*. The egocentric y and z axes are marked in green and blue respectively. The head aim vector (x-axis) is going into the plane of the paper and is denoted by a red circle.

## Error analysis of the 3D head-aim reconstruction

As the dimensions of the headstage were known and remain fixed over the period of the experiment, tracking errors due to the motion capture system is simplified. For example, the distance between the P and Q head markers was 21 millimeters (see *Figure 2—figure supplement 1B*). We allowed a maximum error of 1 millimeter. Tracked points that exceeded this error threshold were excluded from the analysis. In reality, the error in distance between markers is actually a distributed error in the position of the two markers (P and Q in this case). We show this error as grey spheres/discs around each marker in *Figure 2—figure supplement 1B*. The head-aim is reconstructed as the vector $\overrightarrow{PM}$. To compute the maximum and average error in the estimation of the head-aim vector, it is important to estimate the error in computing the midpoint of $\overline{QR}$. We compute this error by first estimating the errors in the coordinates of M.

For simplicity, let us consider a 2D case and let M be the origin as shown in *Figure 2—figure supplement 1C*. Hence, the coordinates of Q and R can be written as (-L, 0) and (L, 0), respectively. Where, 2L is the length of $\overline{QR}$. Let us consider points $Q'(x_{Q'}, y_{Q'})$ and $R'(x_{R'}, y_{R'})$ which belong to the circles of radius 'r' centered at Q and R, respectively and point $M'(x_{M'}, y_{M'})$ which is the midpoint of

$\overline{Q'R'}$. Here 'r' is the maximum allowed error in distance estimation of $\overline{QR}$ (See *Figure 2—figure supplement 1C*). Equations of circles can be written as below (8)

$$(x_{Q'} + L)^2 + y_{Q'}^2 \leq r^2 \text{ and } (x_{R'} - L)^2 + y_{R'}^2 \leq r^2 \tag{8}$$

Adding these equations and rearranging the terms we can rewrite the final equation as

$$\left(\frac{(x_{Q'} + x_{R'})}{2}\right)^2 + \left(\frac{(y_{Q'} + y_{R'})}{2}\right)^2 \leq (x_{M'}^2 + y_{M'}^2) \leq \frac{r^2}{2} - \frac{L^2}{2} + \frac{\left|\overrightarrow{MQ'}\right|\left|\overrightarrow{MR'}\right|\cos\alpha}{2} \tag{9}$$

Where $\alpha$ is the angle between the vectors $\overrightarrow{MQ'}$ and $\overrightarrow{MR'}$ as shown in *Figure 2—figure supplement 2C*. Solving the equation for the extreme cases when $\alpha$ is 0 or 180 degrees shows that *equation (9)* reduces to (10) proving that the error in the estimation of the midpoint M' is also a sphere/circle of radius 'r'.

$$x_{M'}^2 + y_{M'}^2 \leq r^2 \tag{10}$$

*Figure 2—figure supplement 1D* shows the head-aim vector as $\overrightarrow{PM}$ and the grey circles around each point as the error in the position of each marker. In the 2D case, as shown in *Figure 2—figure supplement 1D* it is easy to prove that the maximum angular error in the estimation of the head-aim vector is the angle between $\overrightarrow{PM}$ and $\overrightarrow{T_1 T_2}$, where $\overrightarrow{T_1 T_2}$ is the line tangent to both maximum error circles (indicated in grey) and is can be computed as given in (11).

$$\beta_{err(max)} = \sin^{-1}\frac{r}{L} = 5.45° \tag{11}$$

## Error analysis of the point object approximation

When estimating echo arrival times and echo source locations, all objects are assumed to be point objects and sources. *Figure 2—figure supplement 2A* shows the cross-section of a cylindrical object, which was used as an obstacle in the bat's flight path. The error in the estimation of echo arrival time depends on the position of the bat with respect to the object. *Figure 2—figure supplement 2B* shows how the error in estimation of echo arrival changes as a function of the angle ($\theta$) between the bat's position and the object's horizontal axis, as shown in *Figure 2—figure supplement 2A*. *Figure 2—figure supplement 2C* shows a computation of the accuracy of the echo model as a function of the position of the bat as it moves around the object in a sphere of 2 meters. To summarize, the minimum and maximum errors in time of arrival of the echo at the bat's ears, due to the point object approximation are 0.35 milliseconds and 0.68 milliseconds.

## Echo model validation

The echo model was verified using two different approaches, as detailed below.

1. We Broadcast sounds from a speaker and recorded echoes reflected back from objects using a microphone (shown in *Figure 2—figure supplement 2D*). Here, the distance to the object from the microphone/speaker is 'd' while 'L' is the distance used by the echo model due to the point object approximation. This introduces a systematic error of 'L-d' in the time of arrival of the echo. In this setup the reflecting object was placed at different distances from the speaker and microphone and recorded echo arrival times were compared with the arrival times computed by the echo model. *Figure 2—figure supplement 2E* shows spectrograms of microphone recordings when the object was placed 0.7, 1.2 and 1.8 meters away from the recording microphone. The results matched the theoretical error bounds (as discussed above and shown in *Figure 2—figure supplement 2A,B and C*) within an error less than 0.1 milliseconds (*Figure 2—figure supplement 2F*).
2. A 14-channel microphone array was placed on the wall opposite to the flight direction of the bat. As the bat navigated around objects in its flight path, the microphone array recorded echoes reflected off of objects. Using Time of Arrival of Difference (TOAD) algorithms (*Madsen and Wahlberg, 2007*), the 3D locations of the echo sources were computed and

matched with the locations computed by the *echo model* (see supplementary video SV1 and SV2).

## Classification of neurons into sensory, sensorimotor and vocal-premotor cells

In order to classify neurons, we developed an algorithm based on variability in the firing latency distributions of spike times with respect to echo arrival time, previous call production time, and next call production time. In simple terms, this algorithm measures the variability in spike latencies to echo time and call time (previous and next) as a way of classifying neurons as sensory, vocal premotor or sensorimotor. This determination was based on the assumption that a neuron's activity is most temporally coupled with its functionally relevant event. If a neuron's spike latency distribution was sharpest with respect to echo arrival time, it was classified as sensory; if spike latencies were sharpest with respect to pulse time, the neuron was classified as vocal premotor, and if spike latencies showed clustering around pulse time and echo arrival times, it was classified as sensorimotor. It is important to mention that for sensory neurons we further solved the problem of echo assignment by only considering neurons that fire for the first arriving echo and do not exhibit activity for subsequent echo events (see *Figure 3*). This also solves the problem of wall/camera/microphone echoes, as they were the last to arrive. More than 90% of the sensory neurons analyzed in this study responded only to the first echo. For the remaining neurons that responded to a cascade of echoes (about 10% of those sampled), it was not possible to reliably assign their activity to specific echo arrival times and we therefore excluded them from the data reported in this paper. Using this algorithm, the 182 recorded neurons were classified as sensory (n = 67), vocal premotor (n = 26) and sensorimotor (n = 45). Classification into sensory, sensorimotor and premotor categories is common for SC neurons (*Mays and Sparks, 1980*; *Schiller and Koerner, 1971*). The remaining 44 neurons were unclassified. Spatial tuning profiles were only constructed for the sensory neurons (n = 67).

## Construction of spatial response profiles

Once a neuron was identified as sensory (see above criterion), direction information from the echo model was converted into egocentric coordinates of the bat's instantaneous position and the X, Y and Z information was converted into azimuth, elevation and range coordinates. Further, we test spatial selectivity based on an ANOVA ($p < 0.05$) performed along each dimension (azimuth, elevation and range). Only cells which passed the ANOVA for each dimension were used for further analysis. Neural responses of cells that passed the spatial selectivity test were normalized based on the amount of coverage in each of these dimensions, as explained below.

The spatial response profiles (for neurons which pass the spatial selectivity test (see above) were then normalized using the stimulus space, that is, the time spent by the animal, in each dimension (see *Figure 3—figure supplement 1D* – range, E – azimuth and F – elevation): that is, the spike-count spatial response profile was divided by the time-spent spatial profile, to yield a spiking probability per bin in each dimension (distance, azimuth, and elevation). Regions of the stimulus space with echo events per bin less than one standard deviation from the mean were excluded from the computations (indicated by open bins in *Figure 3—figure supplement 1D, E and F*). Finally, normalized spatial response profiles in each dimension were then fit to a Gaussian function using the *fit* function in Matlab. Spatial response profile means, half widths and standard deviations are then taken from the Gaussian fit.

Out of the 67 sensory neurons (see criterion above), overlapping populations of neurons showed either 3D, 2D or 1D spatial selectivity. 46 neurons (Bat 1–19 and Bat 2–27) showed spatial selectivity in 3D (azimuth, elevation and depth). Further, 56, 52 and 51 neurons showed 1D spatial selectivity, for depth, azimuth and elevation, respectively. *Figure 4—figure supplement 3* describes the complete distribution of 3D, 2D and 1D neurons. The mean response latencies of single sensory neurons we recorded was $5.9 \pm 3.4$ ms. In more detail, the minimum spike latency was 3 ms and the minimum s.d. of latency was 1 ms. The median s.d. of the response latencies for the 67 sensory neurons was 3.8 ms. Previous publications have reported a wide range of response latencies in the SC of the passively listening bat, as long as 40 ms, but also as short as 4 ms (*Valentine and Moss, 1997*), 3.6 ms (*Jen et al., 1984*) and 4 ms (*Wong, 1984*), and short latency responses are likely mediated through a direct projection from the nucleus of the central acoustic tract to the SC (*Casseday et al., 1989*).

## Stability of 3D spatial receptive fields

Further, we determined the stability of receptive fields for individual 3D tuned neurons (n = 46) by comparing the spatial tuning for the first and second half of recording sessions. 37 neurons showed significant 3D spatial selectivity for both the first and second half (see above methods for details). Firing is sparse in the auditory system of echolocating bats, we believe that because of this sparse firing, nine neurons (out of 46) did not show significant spatial tuning (in either the first or second half of the recording session) as a result of limited amount of data in either the first or second half of the recording session. On comparing the selectivity for the first and second half of the recording session, 33 neurons did not show any change in peak tuning along any dimension. Only four neurons showed a significant change in tuning across the session (two in the distance dimension and one each in azimuth and elevation dimensions), thus demonstrating that a majority of the neurons have stable receptive fields across the recording session. *Figure 4—figure supplement 1*, shows the stability of spatial tuning for the depth dimension. Red dots indicate neurons that show a significant change in depth tuning across the first and second half of the recording session.

Neural selectivity was analyzed only with respect to spatial selectivity along the X, Y, and Z dimensions. The bat's echolocation calls are wide-band frequency modulated sounds, which are well suited to evoke activity from SC neurons that respond well to broadband acoustic stimuli. Since variations in the bat's own calls evoked echoes that stimulated SC neurons, we could not systematically analyze responses to other stimulus dimensions, such as sound frequency or intensity. Stimulus selectivity of SC neurons in the bat to non-spatial acoustic parameters will be the topic of a future study.

## SSG and non-SSG analysis

Separate range tuning profiles are computed for each cell for SSG and non-SSG vocalizations. Variance (sharpening) of SSG and non-SSG tuning profiles was tested using the non-parametric Brown-Forsythe test of variance at the α level of 0.05. The test results for each cell are described in detail in table supplementary table 1 (also see *Figure 5E*). Also, SSG and non-SSG distance tuning curves were tested using the Wilcoxon rank-sum test. Test statistic details for each cell is given in table supplementary table 2 (also see *Figure 5F*).

## Power analysis of sample sizes for the SSG and non-SSG spatial tuning comparisons

The firing of auditory neurons in the echolocating big brown bats is very sparse (see for example *Dear et al., 1993*; *Valentine and Moss, 1997*). For the SSG and non-SSG analysis (above) we separated spiking activity when the bat produced SSGs and nonSSGs. This resulted in some of the data sets containing low spike counts. To ensure that for each comparison, for each neuron, we had enough statistical power, we performed a permutation test. Here, we combined the data for SSG and nonSSG data sets and randomly shuffled and picked spikes (without repetitions). Following this, we performed the Brown-Forsythe test or the Wilcoxon rank-sum test, for the sharpening and shifting groups, respectively. We repeated this procedure 1000 times and each time we collected the value of the test statistic. Finally, we compared the test statistic value of the original sample to the distribution obtained from the shuffled groups and obtained a p-value. We only included in the analysis the cells, which passed the test at the $p < 0.05$ criterion level, which excluded 3/56 cells from *Figure 5E* and 5/56 cells from *Figure 5F*.

## Local field potential

The local field potential (<300 Hz) was extracted from each channel recording using second order elliptical filters. Further, we analyzed the gamma band (50–140 Hz) (*Goddard et al., 2012a*; *Sridharan and Knudsen, 2015*) to investigate whether the epochs when the bat produced sonar sound groups (SSGs) were correlated with gamma band activity. We first identified channels without distortions in the LFP as a result of movement artifact (*Figure 6—figure supplement 2*). We then extracted 100 ms spike triggered LFP windows from corresponding recording sites. We separated these into SSG and non-SSG events and averaged these separately to estimate the root mean squared (RMS) gamma band power (*Jaramillo and Zador, 2011*) (*Figure 6A and B*) when the bat produced SSG and non-SSGs. Further, to investigate the timing of the gamma signal, the averaged gamma band amplitude envelope was normalized across SSG and non-SSG trials across each

neuron. A Gaussian was fit to each time waveform to estimate the peak (*Figure 6C and D*). The average of the peaks across all units was taken as the average latency of the LFP following the spike event.

We also examined whether movement artifact from the bat's wing beats could have corrupted the LFP analysis. The bat's wingbeat is approximately 12 Hz, whereas the frequency range for the Gamma band we analyzed was 50–140 Hz. The third harmonic of the wingbeat, which would be in the frequency range of the Gamma band, was significantly attenuated. To further ensure that movement artifact did not corrupt the analysis of the LFP, we chose channels where the power ratio between the low frequency band (10–20 Hz) and the gamma band was less than 6 dB. We identified 21 low noise channels containing 26 single neuron recordings, (see *Figure 6—figure supplement 2*), which were then used for further analysis.

### Data and code availability

The original raw data can be obtained upon request from NBK, MJW or CFM (cynthia.moss@jhu.edu). Given the size of the raw data (approx.. 2 terabytes), the full dataset has not been deposited to a public repository, but partial and processed data sets to generate *Figures 5E, 5F*, *6C, 6D and E* have been made available through an open source license on GitHub (*Kothari et al., 2018* copy archived at https://github.com/elifesciences-publications/Dynamic-3D-auditory-space-in-bats).

## Acknowledgements

We would like to thank Drs. Nachum Ulanovsky, Yossi Yovel, Uwe Firzlaff, Lutz Wiegrebe and Shreesh Mysore for comments on the research presented in this manuscript. We also thank the members of the Johns Hopkins Comparative Neural Systems and Behavior lab (aka Bat Lab) for their valuable feedback on data analysis reported in this article, Dallas DeFord for help with data preprocessing, and James Garmon of Psychological and Brain Sciences, JHU for designing and fabricating various apparatus used during data collection, without which this experiment would not have been possible. This work was supported by the following research grants NSF IOS1460149, AFOSR FA9550-14-1-039, ONR N00014-12-1-0339 and ONR MURI N00014-17-1-2736.

## Additional information

### Funding

| Funder | Grant reference number | Author |
|---|---|---|
| National Science Foundation | IOS1460149 | Cynthia F Moss |
| Air Force Office of Scientific Research | FA9550-14-1-039 | Cynthia F Moss |
| Office of Naval Research | N00014-12-1-0339 | Cynthia F Moss |
| Office of Naval Research | N00014-17-1-2736 | Cynthia F Moss |

The funders had no role in study design, data collection and interpretation, or the decision to submit the work for publication.

### Author contributions

Ninad B Kothari, Conceptualization, Data curation, Software, Formal analysis, Validation, Investigation, Visualization, Methodology, Writing—original draft, Writing—review and editing; Melville J Wohlgemuth, Conceptualization, Data curation, Validation, Investigation, Writing—original draft, Writing—review and editing; Cynthia F Moss, Conceptualization, Resources, Supervision, Funding acquisition, Project administration, Writing—review and editing

### Author ORCIDs

Ninad B Kothari (iD) http://orcid.org/0000-0001-5543-6459
Melville J Wohlgemuth (iD) http://orcid.org/0000-0003-4779-4154
Cynthia F Moss (iD) http://orcid.org/0000-0001-6916-0000

### Ethics

Animal experimentation: All of the animals were handled according to approved institutional animal care and use committee (IACUC) protocols of the Johns Hopkins University. The experimental protocol (BA17A107) was approved (March 16, 2017) by the IACUC of the Johns Hopkins University. All surgery was performed under isoflurane anesthesia, and every effort was made to minimize suffering.

### Decision letter and Author response

Decision letter https://doi.org/10.7554/eLife.29053.027
Author response https://doi.org/10.7554/eLife.29053.028

## Additional files

### Supplementary files

• Supplementary file 1. (**A**) Comparison of the variance of SSG and non-SSG distance tuning distributions for each cell in *Figure 5D*. The SSG and non-SSG distance tuning distributions were compared using the non-parametric Brown-Forsythe Test at the level α of 0.05. Cells in red show a significant sharpening in the distance tuning distribution when the bat emitted SSGs as compared to the variance of the distance tuning distribution when the bat produced single calls (non-SSGs). Cells in gray did not show a significant effect. Cells in blue showed a significant effect but in the opposite direction. (**B**) Comparison of the SSG and non-SSG distance tuning distributions for each cell in *Figure 5F*. The SSG and non-SSG distance tuning distributions were compared using the non-parametric Wilcoxon Rank Sum Test at an α of 0.05. Cells marked with red ink show a significant shortening in distance tuning for SSGs as compared to the condition when the bat produces single calls (non-SSGs). Cells marked with gray ink did not show a significant effect.
DOI: https://doi.org/10.7554/eLife.29053.024

• Transparent reporting form
DOI: https://doi.org/10.7554/eLife.29053.025

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
