## [Decision Letter]

Thank you for submitting your article "Dynamic representation of 3D auditory space in the midbrain of the free-flying echolocating bat" for consideration by *eLife*. Your article has been favorably evaluated by Andrew King (Senior Editor) and three reviewers, one of whom is a member of our Board of Reviewing Editors. The reviewers have opted to remain anonymous.

The reviewers have discussed the reviews with one another and the Reviewing Editor has drafted this letter to assess your responses to the concerns raised by the reviewers.

Summary:

The authors have developed a telemetry recording device mounted on a big brown bat and recorded responses from neurons in the superior colliculus to the bat's own echolocation calls in free flight. This is a major finding, requiring use of a light weight telemetry device, and demonstrating delay-tuned neural responses to echoes from physical objects. Previous work had used simulated pulses and echoes. Other significant results are the demonstration that echo clusters are correlated with sharpened responses.

Essential changes:

Despite these achievements, there are some major concerns. Data are reported from only 2 bats, and only 41 single units are selective to pulse-echo pairs. These are not described in sufficient detail to evaluate the results. Detailed questions are provided with the individual reviews, which are provided below. How can the authors show that the bats were actually using their sonar system for performing the task at hand and hence attending and actively interpreting those signals? It is not clear whether the responses were recorded from bats prior to training to fly around the room, or during, or after training. How many days did each bat wear the telemetry device? Were neurons recorded when the bat was passively listening to sounds? Could echoes ever be recorded in order to precisely determine echo-response latencies?

The second major issue concerns the model, which has been used to replace recording of echoes. This is described as technical advance, and appears to be fundamental to the analysis of neural responses. Nevertheless, insufficient details are provided to validate the model. Your flight room was lined with acoustic foam to reduce multiple echoes from the surroundings; however, surely the bat must have known the position of the walls etc. from its own echolocation? Were there (maybe faint) echoes from the floor or the camera/microphones?

When referring to the sharpening of the neuronal response and the shift of the response towards closer targets during grouped pulses you emphasize the potential role of attention, and attention-guided sonar pulse emission. You also argue that neuron's best delay (target distance) is shifted to shorter delays when the bat is engaged in the production of SSGs. A detailed analysis of the neural responses of single neurons during SSGs across different locations in the room would be important for accurately assessing the observed correlated changes in neural activity.

The last major concern is that the authors clearly delineate the novelty of this study, since this was questioned by the reviewers.

Please also respond to the remaining reviewer concerns below.

Reviewer #1:

It is not clear whether the authors ever record the echoes that lead to their echo-evoked spikes, or just modelled them. The neural responses are described as responses to sensory locations in 3D egocentric space, but I cannot see the echoes on the figures.

Some of the claims of novelty seem overblown. The transformation from 2D to 3D spatial receptive fields is inherent in bat echolocation, and neurons tuned to different spatial locations and ranges have been described multiple times.

Reviewer #2:

In this study neuronal activity in the superior colliculus was recorded in free flying bats that have a wireless-multielectrode system implanted. Methodologically this study provides a breakthrough: For the first time it was possible to study spatial neuronal coding in an echolocating bat while the animal was freely moving and was confronted with several objects in its flight room. To be able to associate neuronal space coding with instantaneous occurrence of sonar echoes at the bat's ear, the authors employ a sophisticated model for predicting echo occurrence at the bat's ears that includes bat flight trajectory and head position filmed by several cameras, bat echolocation calls measured with a microphone array, and estimates of sonar beam width. This elegant technique is impressive and avoids having to put microphones on the bats' heads and it produces novel results: The authors convincingly show that the spatial focus of SC neurons sharpens and shifts to closer objects when the bat employs sonar sound groups, e.g. call doublets, to inspect its environment. The latter are indicative of increased attention. The Discussion could benefit from coverage of literature on sharpening and shift of spatial neuronal tuning in other auditory areas like the auditory cortex.

The study is very well done, it employs sophisticated new methods and its novel results clearly merit publication in *eLife*.

Reviewer #3:

The proposed manuscript by Kothari et al. describes their investigation concerning encoding properties of midbrain neurons for egocentric space from a freely moving animal. By doing so in free flying bats navigating via their echolocation system they could correlate the neural activity with the bats auditory responses to ongoing echo-sampling of the environment. Through the aid of a physics-based echo-model the authors revealed that not only do sensory neurons in the superior colliculus encode 3D egocentric space but they also appear to sharpen their tuning curve with respect to the bat ongoing behavior. The authors' finding support the hypothesis that a bat's echolocation system could enhance auditory spatial resolution comparable to how eye saccades and foveal fixation enhances it for the visual system. Their results show for the first time how sensory neurons sharpen their 3D spatial resolution during a natural spatial navigation task. While the results are exciting from that perspective there are many details that need to be addressed both at the level of the behavior as well as the level of the neural data in order to determine whether the presented data supports the authors conclusions. Furthermore, a more refined description of the novelty of this study would be helpful.

The description of the behavior, task and performance is severely lacking which hinders accurate assessment of the authors interpretation of the neural data. Considering the well-established modulation of fine behavioral parameters, training procedures and sensory input on neural activity in the superior colliculus (SC) these details carry a lot of weight in validating the conclusions proposed by the authors. A few notable examples are the following:

1) How can the authors show that the bats were actually using their sonar system for performing the task at hand and hence attending and actively interpreting those signals?

2) How reproducible was the flight behavior of the bats? If it was highly stereotyped could it be that the bat was relying more on spatial memory and hence only attending partially to a small fraction of the echo-pulses as "verification" sampling? If this is indeed the case then could it be that the observed differences by the authors are not a dynamic modulation of the spatial signal but more so a transition between different behavioral states. It would also be informative if the authors report the number of trails that go into each analysis, the spatial density of the flight behavior, as well as show all the three-dimensional flight paths from representative sessions so a better assessment of the animal's spatial behavior can be obtained.

3) How were the animals trained on this task and how well trained were the bats? Is it possible that the sharpening of the neural spatial resolution is dependent on how well a task is known, meaning a high expectancy of animal? Did the authors observe any changes in the sharpening on dynamic tuning of the neural activity during the days of poorer vs. better performance?

4) What was the reward contingency? Neural activity in SC is believed to be modulated by upcoming reward (see for example Ikeda and Hikosaka, Neuron, 2003). Did the authors observe different patterns of neural activity near rewarded (for example landing platform) vs. non-rewarded (for instance hanging obstacle)? Was the location of the obstacles changed between recording sessions and did this manipulation had an effect on the animal behavior and neural activity?

5) Can the authors show that the bats used echolocation rather than vision to perform the task? I assume the room was dark during the experiment but how dark was it (lux levels)? Can the authors show that the bat behavior was modulated by the echo-pulses or alternatively, that they did not use vision for attending to distal cues and echolocation for attending to more proximal cues (which might provide an alternative explanation for neural responses)? This information is important in order assess the extent to which these bats used different sensory modalities (audition vs. vision) in performing different aspects of this task. As a side note, most often high-speed video tracking systems employ infrared-red lighting. Is there data for *Eptesicus fuscus's* wavelength sensitivity?

6) Can the authors provide a more detailed description as to the type/size/shape of the objects used in their task? The results seem to show that the bats only needed a glimpse of the object and used this mainly to detect the objects' spatial positions, not needing more information of the spatial extent of an object. Was this indeed the case for most objects presented or were some inspected for longer durations of time? If so, did this influence the echolocation signals and corresponding neural activity?

7) What echolocation signals are actually being analyzed here? Is it only to returning echoes or also to "missed" echoes, meaning did the authors also look at all neural responses after an echolocation call was made independent of whether this resulted in a reflection or not?

In addition to the analysis of neural data with respect to behavior noted above, further basic information should be analyzed and provided by authors to better assess the quality of the presented data and its validity for the authors interpretation. For instance:

1) What was the quality of the sorting? How well were the clusters separated into single units? The authors should provide quantitative measures for the quality of their sorting (such as isolation index and L_ratio_).

2) Furthermore, the authors describe a threshold of 200 spikes for a cell to be included in the analysis. How was this threshold chosen and does this number only include spikes recorded during flight? The authors further set a seemingly arbitrary threshold of at least 20 spikes during SSG and non-SSG which results in a total sample of 20 neurons for this entire analysis? If this is indeed the case it seems rather concerning basing conclusions on 20 action potentials and such a low number of neurons. The authors should either increase their neuron count or show statistically the strength of their threshold as valid for avoiding low sample biases for this analysis.

3) How were the motion artifacts shown by the authors in Figure 1—figure supplement 1 characterized and corrected? How were these artifacts detected and were they constrained to more specific spatial locations (such as the landing platform) or equally distributed throughout the flight paths? Could these artifacts obscure the detection of action potentials in certain parts of the environment more than in others? What did the authors do to address the potential influence of such artifacts on the neural signals and especially for on LFP signal? For instance, motion artifacts can cause the distortion of electrical signals in low frequency bands. To what extent were gamma oscillations influenced by such interferences? If any patterns of wing motions were associated with SSGs then this could account for stronger fluctuations in LFP hence seemingly increasing the gamma power.

4) What part of the frequency band of gamma was analyzed? The authors define a very wide band (40-140Hz) but there is a clear distinction between low and high gamma frequency bands that should also be analyzed. Furthermore, did the authors observe differences both at the gamma band effect as a function of the recorded layer in the superior colliculus? (see for example the paper by Ghose, Maier, Nidiffer and Walace, Multisensory response modulation in the superficial layers of the superior colliculus, Journal of Neuroscience, 2014).

5) Did the selectivity of the neurons change as a function of anatomical location of the recorded neurons within the SC (such as more superficial vs. deeper layers of SC)?

6) Stability: Figure 1—figure supplement 1 the authors show some action potentials from a single session yet this does not address in a meaningful way the question of whether the neural features observed by the authors represent a stable phenomenon related to the bat echolocation alone or is it actually modulated by different factors which are changing during the task. For such analysis, the authors should compute the stability of the neural responses of all the analyzed neurons during equally spaced portions of each session and show quantitatively that the response profiles (such as tuning curves) remain stable throughout.

7) The definition of neural selectivity in unclear. What is the formal definition by which a neuron was determined to be selective along a particular dimension? Can the authors provide a formula and statistical description for how a neuron was selected for the analysis (the paper is primarily relying on just 41 sensory selective neurons)? How were sensory, sensorimotor and vocal-premotor neurons classified? How many were recorded from each bat? What statistical threshold was used to determine selectivity? Furthermore, was the selectivity of a neuron analyzed along any orthogonal set of dimensions which are different from the canonical x, y and z dimensions?

8) The authors argue that neuron's best delay (target distance) is shifted to shorter delays (closer objects) when the bat is engaged in the production of SSGs, suggesting that distance tuning is dynamically remapped when the bat actively inspects objects in its environment. So, what actually drives the observed change in neural activity? Is it the result of higher echolocation rates? Is it that the bat simply in a different behavioral mode? Is it the physical proximity of objects? A detailed analysis of the neural responses of single neurons during SSGs across different locations in the room, either near or further away from the target and at different echolocation rates would be important for accurately assessing the observed correlated changes in neural activity.

Lastly, while the animal model and experimental preparation are certainly exciting, the paper in its current form does not provide revolutionary new insight nor a methodological innovation beyond what has already been reported in previous studies. Neural recordings have already been done in freely behaving and flying bats and the dynamic encoding of spatial locations and attention by both single neurons and LFP (gamma oscillations in this case) have already been reported in a wide range of species. Yet, this paper does provide an important confirmation of previous hypothesis and data in the freely flying bat and for that it does provide important insight.

In detail, the fact that perceptual sensitivity of SC neurons changes in relation to spatial cues is not novel has already been shown previously in behaving animals (See for example Lovejoy and Krauzlis, 2017). Albeit in some cases in non-human primates the head is restrained, they are performing active sensing using their eyes which are freely moving. The authors further argue that most studies have studied only 2D spatial cues but the three-dimensional tuning properties of neurons in SC have also been shown previously in the same bat species in the same lab (Valentine and Moss, 1997), albeit this was not done in freely flying bats. A dynamic modulation of neural activity with respect to the bat echolocation signals have also been previously demonstrated both during behavior in the same species (Ulanovsky and Moss, Hippocampal, 2011) as well as during flight in a different species (Geva et al., Nature neuroscience, 2016). Yet these recordings were done in the hippocampus and not in the SC as in the present study. Hence, the importance of this work is that it brings many of these components together in the superior colliculus of the flying bat. This aspect is important as it allows addressing long standing hypotheses about the tuning properties of such neurons in echolocating bats. It is important however that the authors properly and clearly delineate the statements of novelty in this study. As a side note: The authors further make the statement in the Introduction that "past studies of the SC have largely focused on sensorimotor representation in restrained animals, leaving gaps in our knowledge about the influence of action and attention on sensory responses in freely moving animals". Yet this statement is also not entirely accurate. Studies in rodents have also provided important contribution on the role of sensory input for modulating action and neural activity in the SC and should be acknowledged (see for example Felsen and Mainen, 2008.

[Editors' note: further revisions were requested prior to acceptance, as described below.]

Thank you for resubmitting your work entitled "Dynamic representation of 3D auditory space in the midbrain of the free-flying echolocating bat" for further consideration at *eLife*. Your revised article has been favorably evaluated by Andrew King (Senior Editor) and three reviewers, one of whom is a member of our Board of Reviewing Editors.

The manuscript has been improved but there are some remaining issues that need to be addressed before acceptance, as outlined below, and with more details provided in the individual reviews. All of these points relate to the analysis and presentation of the data, and the reviewers agreed that the novelty and impact of this study will be clearer once this is done.

1) Quantification of the behavior (represented either by correlation of flight trajectories, heatmaps or a different method).

2) Since there are data from only two bats, perhaps the authors could show the distribution of the main results across those two bats (as one would do for non-human primates which often have two subjects) to allow the readers to compare the responses.

3) Provide the standard measurement of unit-isolation quality by showing the distribution of L_ratio_ and isolation distance for all of the analyzed neurons and exclude neurons which are clearly multi-unit. The concern is that biases can emerge from noisy analysis relying on very low spike counts. The authors should provide a measure of what is a reasonable threshold of number of spikes for a single neuron such that it can be safely included in the different analysis, or show that their results do not depend on spike counts.

Reviewer #1:

The authors have been very responsive to my previous review. I accept their reasons for relying on their model for the timing of echo-evoked spikes.

With respect to neurons recorded, although data are still from only 2 bats, they have increased the number of neurons analyzed. The authors now report 182 single neurons recorded in the SC, with 67 being selective to pulse-echo pairs. The authors have answered questions about unit isolation with a citation to the Quiroga et al. 2004 paper, and also point out that data from both bats show similar results. More information should be provided about these units. The authors could provide data on their units to support divisions into sensory, motor, etc., including latency and rate, as well as spike sorting criteria.

In the revision, the authors provide more data on their SSG responses. It would be helpful to find out how the 26 units shown for the SSGs were classified? The only group of neurons that I can find with n=26 are the vocal premotor group. Are these the same units?

The other revisions greatly add to the paper, providing data on bat flight paths, and number of sessions in which responses were recorded.

Reviewer #2:

The manuscript has improved, the authors took into account all my questions/comments in their new manuscript. The manuscript could be published as it is.

Reviewer #3:

While I appreciate the response from the authors I still find the information provided lacking on the three main domains I described in my original review:

1) Behavioral data analysis and presentation: Still there is no detailed description and importantly, quantification of the behavior. The authors say that bats were released from different locations and did not exhibit stereotyped behaviors but as requested, they should show this. As requested previously, can the authors provide a quantitative assessment of the reproducibility (or lack-thereof) of both the starting positions and flight trajectories. This can, for example, be done in the form of a heat-map illustration the spatial distribution (in 3D space) of the flight trajectories. But I encourage the author to provide a different assessment of this important point as well such that their argument for more variable flight trajectories and starting positions are better supported.

Also, the information on reward contingency is lacking. How many trials in a session resulted in landing on a platform and how many did not? When the bats did not land on the platform were they also rewarded? Much more details of the behavior are required and at the moment not provided by the authors. Again, this information is important for addressing the nature of the neural responses as postulated by the authors and align their results with findings from other species (such as primates, rodents, etc…). This will allow the authors to engage a broader audience beyond the single species of bats.

2) Analysis of neural data: Despite the request in the previous round of reviews there is no quantification of the quality of the neural signal. As requested, the authors should provide some assessment of the quality of their neural data in the form of isolation indexes and L_ratio_. Again, such measures are fairly standard in neural analysis and would allow the readers to assess the data more properly. Also, the authors have now removed the threshold on the minimal number of action potentials for a neuron to be included in their analysis. This allowed them to increase the N of "valid neurons" without a quantification of the dependence of the sensitivity of the observed tuning curves on the number of action potentials included in the analysis. Furthermore, this was not requested by the reviewers and bring about concerns regarding conclusions being made based on very low number of spikes and without support that this cannot bias the result. I encourage the authors to reinstate the threshold and importantly, provide a quantitative threshold that would assure the results are not biased by low numbers of action potentials included in the analysis.

3) Novelty: The points raised by the authors are still in agreement with the fact that this paper very elegantly puts together pieces of data that have been previously reported elsewhere, either in bats or in other species and mostly serve as a verification of previous findings. Furthermore, some of the statements made by the authors are unclear. For example, when comparing to the work of Geva et al., the authors claim that the Egyptian bat does not dynamically modulate its sonar signal in response to its surrounding but the senior author of the paper is an author on a paper stating that it does (S. Danilovich, A. Krishnan, W. J. Lee, I. Borrisov, O. Eitan, G. Kosa, C. F. Moss, Y. Yovel (2015) Bats regulate biosonar based on the availability of visual information Current Biology 25(Feng et al., 1978), 1107-1125), which is puzzling. Furthermore, other studies from Yosef Yovel and Nacham Ulanovsky have demonstrated that this species does in fact dynamically change the directionality of its sonar beam in responses to the acoustic features of its environment, and the senior author of the current paper is also an author on that manuscript: Y. Yovel, B. Falk, C. F. Moss, N. Ulanovsky, (2011) Active control of acoustic field-of-view in a biosonar system, PLoS Biology, 9(Bichot et al., 2005): e1001147. (Open article).

In summary, while I am generally supportive of this important work I still feel that the authors should provide more detailed responses to the requests of the referees and frame their work better with respect to the vast knowledge on the neurophysiological properties of SC neurons across species. The latter would only benefit the authors as it will allow them to extend the impact and interest in their work to a broader audience, such as the readers of e*Life*.

---

## [Author Response]

Essential changes:Despite these achievements, there are some major concerns. Data are reported from only 2 bats, and only 41 single units are selective to pulse-echo pairs. These are not described in sufficient detail to evaluate the results. Detailed questions are provided with the individual reviews, which are provided below.

In our revised manuscript, we provide more details on the numbers of units, and the proportion of the total neurons that were sensory, sensorimotor, and motor. Since our original submission, we have substantially increased the neuron count. We now have data on a total of 182 neurons. Further, after classification (see Materials and methods subsection “Classification of neurons into sensory, sensorimotor and vocal-premotor cells” for details), we now have analyzed data from 67 sensory (auditory) neurons, 45 sensorimotor neurons, and 26 vocal premotor neurons. Out of the 67 sensory neurons, 46 showed 3D spatial receptive fields, the remaining 17 neurons were a mix of 2D and 1D spatially tuned neurons (details are included in the subsection “Construction of 3D spatial response profiles”, also, see Figure 4—figure supplement 2). Further, we would like to emphasize that data from both bats in the study show similar results.

How can the authors show that the bats were actually using their sonar system for performing the task at hand and hence attending and actively interpreting those signals?

In our experiments, the bats’ behavior was modulated by echo returns, and this is indicated by the adaptive changes in sonar call parameters with distance to objects (see Figure 1), in line with findings reported in previous studies (Griffin, 1958; Simmons et al., 1979). The bats also produced sonar sound groups, clusters of echolocation calls, which are observed when bats inspect objects in space (Falk et al., 2014; Moss et al., 2006; Petrites et al., 2009; Sändig et al., 2014; Wheeler et al., 2016).

Additionally, the room lighting was outside the bat’s visible range: ERG data show that *Eptesicus* does not see wavelengths longer than 600 nanometers (Hope and Bhatnagar, 1979). Moreover, the bats were released at different locations in the room for each trial, which required them to use sonar echoes to steer around obstacles rather than a consistent or memorized flight path around objects in the room (also, see (Figure 1) for more details). We have included these details in the revised manuscript (subsection “Experimental design”).

It is not clear whether the responses were recorded from bats prior to training to fly around the room, or during, or after training.

Recordings were taken after the bats had experience in experimental test room. Details have now been added to the text in the Materials and methods section (subsection “Experimental design”).

How many days did each bat wear the telemetry device?

After the chronic implant surgery each bat was given several days to rest and acclimate to the implanted device, after which they were able to fly and navigate around objects in the flight room. Data collection began after the animal was able to perform ~30 flight trials per session, which took place twice a day (morning and afternoon) in the experimental test room. Bat A flew for 12 sessions, and Bat B flew for 15 sessions. These details are now in the text (subsection “Experimental design”, last paragraph).

Were neurons recorded when the bat was passively listening to sounds?

The SC recordings were collected when the bats were navigating, and 3D spatial tuning of SC neurons was constructed off-line with the use of the echo model (see responses to reviewers). Since spatial tuning of neurons was computed off-line, it was not feasible within a 30-45 minute experimental session to compare echo-evoked activity in free-flying bats with passive responses to computer generated pulse-echo pairs constructed to mimic stimulus azimuth, elevation and delay parameters that elicited responses from neurons in flight. We included these details of the methods in the revised manuscript (subsection “Echo model - Reconstructing the instantaneous acoustic stimulus space at the ears of the bat”, second paragraph and subsection “Experimental design”).

Could echoes ever be recorded in order to precisely determine echo-response latencies?

We did not record echoes with a microphone on the bat’s head during the course of the experiment. Below we list reasons why it was not feasible to record echoes directly received at the ears of the free-flying bat to characterize 3D spatial receptive fields.

- Noise floor: The noise floor of microphones is inadequate to record all of the echoes that return to the bat as it vocalizes. This would restrict the construction of spatial receptive fields, as there would be ‘missing echoes’. The echo model overcomes this limitation.

- Single/multiple microphones: A single microphone can only give information about the time of arrival of strong echoes. To obtain echo direction information an array of microphones (at least 4) would be required to use time difference of arrival algorithms for localizing echo sources.

- Size and weight considerations: For bats weighing approximately 18 grams, any increase in the weight of the head mounted devices would seriously limit their flight behavior.

We report in our revision that we used a microphone array to record echoes to validate the echo model’s computation of the arrival time and direction of echoes from objects in the room (see below). We could therefore analyze neural response latencies to the computed arrival times of echoes. The mean response latencies of single neurons we recorded is 5.9 ± 3.4 ms. In more detail, the minimum spike latency was 3 ms and the minimum s.d. of latency was 1 ms. The median s.d. of the response latencies for the 46 sensory neurons was 3.8 ms. Previous publications have reported a wide range of response latencies in the SC of the passively listening bat, as long as 40 ms, but also as short as 4 ms (Valentine and Moss, 1997), 3.6 ms (Jen et al., 1984) and 4 ms (Wong, 1984), and short latency responses are likely mediated through a direct projection from the nucleus of the central acoustic tract to the SC (Casseday et al., 1989).

The second major issue concerns the model, which has been used to replace recording of echoes. This is described as technical advance, and appears to be fundamental to the analysis of neural responses. Nevertheless, insufficient details are provided to validate the model.

We have now included details of the echo model, which were previously in the supplementary material, in the main text. Further, we have elaborated on details of the echo model, as pointed out by the reviewers.

Your flight room was lined with acoustic foam to reduce multiple echoes from the surroundings; however, surely the bat must have known the position of the walls etc. from its own echolocation? Were there (maybe faint) echoes from the floor or the camera/microphones?

The reviewer is correct that the bats received echoes from the walls, especially at closer distances. In the revised manuscript, we have added further details about wall/camera echoes (subsection “Classification of neurons into sensory, sensorimotor and vocal-premotor cells”). Briefly, every call that the bat produced resulted in a series of echoes, which included echoes from ensonified objects, walls, cameras etc. However, due to the close proximity of the bat with respect to the objects, the object echoes were always the first to arrive at the bat’s ears. In our sensory cell classification, we only considered neurons that responded to the first echo from each vocalization, which arrived before wall/camera echoes. These details have been added to the manuscript.

When referring to the sharpening of the neuronal response and the shift of the response towards closer targets during grouped pulses you emphasize the potential role of attention, and attention-guided sonar pulse emission. You also argue that neuron's best delay (target distance) is shifted to shorter delays when the bat is engaged in the production of SSGs. A detailed analysis of the neural responses of single neurons during SSGs across different locations in the room would be important for accurately assessing the observed correlated changes in neural activity.

We have analyzed the bat’s production locations of SSGs and non-SSGs across the experimental flight room for individual sessions and the distributions for both are overlapping and distributed across the room. We would be happy to provide more details of this analysis if required by the reviewers.

We also show in our revised manuscript, that the spatial tuning of cells is stable across each recording session. In this context, we would like to mention that the sparseness in neural activity in the auditory system of big brown bats (Dear et al., 1993; Valentine and Moss, 1997) limits the spiking data we can use for this assessment on a per trial basis. We compared response profiles of neurons in the first and second half of recording sessions, and found that the vast majority of neurons (37 out of 46 neurons) had stable spatial receptive fields between the two halves of each recording session. These details are included in the text in the fourth paragraph of the subsection “Construction of 3D spatial response profiles”, also, Figure 4—figure supplement 1).

The last major concern is that the authors clearly delineate the novelty of this study, since this was questioned by the reviewers.

We have responded, in detail, to this concern by the reviewers (see below) and also list our argument here.

1) 3D sensory responses in a freely behaving bat to self-generated echoes from physical objects have never been previously demonstrated.

2) Our results are the first to show that 3D sensory space is represented in the brain of an animal interacting with objects in its physical environment, and importantly, that sensory representations are modified by adaptive vocal-motor behaviors.

3) Our study is significantly different from Geva et al.’s 2016 report of hippocampal place field remapping in Egyptian fruit bats exposed to different sensory environments (vision vs. echolocation). Not only do the data from the two studies come from recordings in different species and different brainstructures, but also in experiments with different dependent and independent variables.

4) While past studies have implicated the SC in spatial attention and perception through local inactivation (Lovejoy and Krauzlis, 2017; McPeek and Keller, 2004), our work differs in that we recorded activity in single SC neurons in freely behaving animals and demonstrate remapping and sharpening of 3D neurons with changes in the animal’s echolocation behavior.

5) The characterization of 3D response profiles in freely echolocating bats is an important breakthrough. For many decades, scientists have mimicked natural echolocation in restrained (often anesthetized) passively listening bats, without testing the validity of this approach.

6) Importantly, we believe that our research can inspire colleagues to conduct related studies in other species, which would advance a more complete understanding of nervous system function in the context of real-world, natural behaviors.

Please also respond to the remaining reviewer concerns below.Reviewer #1:It is not clear whether the authors ever record the echoes that lead to their echo-evoked spikes, or just modelled them. The neural responses are described as responses to sensory locations in 3D egocentric space, but I cannot see the echoes on the figures.

As stated above, we did not record echoes with a microphone on the bat’s head during the course of the experiment and we explained why it was not possible to record echoes directly received at the ears of the free-flying bat to characterize 3D spatial receptive fields.

- Noise floor: The noise floor of microphones is inadequate to record all of the echoes that return to the bat as it vocalizes. This would restrict the construction of spatial receptive fields, as there would be ‘missing echoes’. The echo model overcomes this limitation.

- Single/multiple microphones: A single microphone can only give information about the time of arrival of strong echoes. To obtain echo direction information an array of microphones (at least 4) would be required to use time difference of arrival algorithms for localizing echo sources.

- Size and weight considerations: For bats weighing approximately 18 grams, any increase in the weight of the head mounted devices would seriously limit the flight behavior.

This information is included in our revised manuscript. We also elaborate in our revision on echo measurements with a microphone array to validate the echo model’s computation of the arrival time and direction of echoes from objects in the room, which allowed us to analyze neural response latencies to the computed arrival times of echoes.

Some of the claims of novelty seem overblown. The transformation from 2D to 3D spatial receptive fields is inherent in bat echolocation, and neurons tuned to different spatial locations and ranges have been described multiple times.

This has been addressed briefly in our response to the Editor’s general comments and in more detail below in response to the comments of reviewer 3.

Reviewer #2:In this study neuronal activity in the superior colliculus was recorded in free flying bats that have a wireless-multielectrode system implanted. Methodologically this study provides a breakthrough: For the first time it was possible to study spatial neuronal coding in an echolocating bat while the animal was freely moving and was confronted with several objects in its flight room. To be able to associate neuronal space coding with instantaneous occurrence of sonar echoes at the bat's ear, the authors employ a sophisticated model for predicting echo occurrence at the bat's ears that includes bat flight trajectory and head position filmed by several cameras, bat echolocation calls measured with a microphone array, and estimates of sonar beam width. This elegant technique is impressive and avoids having to put microphones on the bats' heads and it produces novel results: The authors convincingly show that the spatial focus of SC neurons sharpens and shifts to closer objects when the bat employs sonar sound groups, e.g. call doublets, to inspect its environment. The latter are indicative of increased attention. The Discussion could benefit from coverage of literature on sharpening and shift of spatial neuronal tuning in other auditory areas like the auditory cortex.The study is very well done, it employs sophisticated new methods and its novel results clearly merit publication in eLife.

We thank the reviewer for these comments and have expanded the literature review in the Discussion of our revised manuscript.

Reviewer #3:The proposed manuscript by Kothari et al. describes their investigation concerning encoding properties of midbrain neurons for egocentric space from a freely moving animal. By doing so in free flying bats navigating via their echolocation system they could correlate the neural activity with the bats auditory responses to ongoing echo-sampling of the environment. Through the aid of a physics-based echo-model the authors revealed that not only do sensory neurons in the superior colliculus encode 3D egocentric space but they also appear to sharpen their tuning curve with respect to the bat ongoing behavior. The authors' finding support the hypothesis that a bat's echolocation system could enhance auditory spatial resolution comparable to how eye saccades and foveal fixation enhances it for the visual system. Their results show for the first time how sensory neurons sharpen their 3D spatial resolution during a natural spatial navigation task. While the results are exciting from that perspective there are many details that need to be addressed both at the level of the behavior as well as the level of the neural data in order to determine whether the presented data supports the authors conclusions. Furthermore, a more refined description of the novelty of this study would be helpful.

We thank the reviewer for pointing out the shortcomings in our presentation of data collection and analysis, and we have made every effort to correct this by adding extensive detail to our Materials and methods and Results sections. We specify our changes below.

The description of the behavior, task and performance is severely lacking which hinders accurate assessment of the authors interpretation of the neural data. Considering the well-established modulation of fine behavioral parameters, training procedures and sensory input on neural activity in the superior colliculus (SC) these details carry a lot of weight in validating the conclusions proposed by the authors. A few notable examples are the following:1) How can the authors show that the bats were actually using their sonar system for performing the task at hand and hence attending and actively interpreting those signals?

We thank the reviewer for pointing out this confusion in our manuscript. In our experiments, the bats’ behavior was modulated by echo returns, and this is indicated by the adaptive changes in vocal behaviors. Additionally, the infrared room lighting was outside the bat’s visible range: ERG data show that *Eptesicus* does not see wavelengths longer than 600 nanometers (Hope and Bhatnagar, 1979). Moreover, the bats were released at different locations in the room for each trial, which required them to use sonar echoes to steer around obstacles rather than a consistent or memorized flight path around objects in the room (also, see (Figure 1) for more details). We have included these details in the revised manuscript (subsection “Experimental design”).

2) How reproducible was the flight behavior of the bats? If it was highly stereotyped could it be that the bat was relying more on spatial memory and hence only attending partially to a small fraction of the echo-pulses as "verification" sampling? If this is indeed the case then could it be that the observed differences by the authors are not a dynamic modulation of the spatial signal but more so a transition between different behavioral states. It would also be informative if the authors report the number of trails that go into each analysis, the spatial density of the flight behavior, as well as show all the three-dimensional flight paths from representative sessions so a better assessment of the animal's spatial behavior can be obtained.

We varied the locations of the four flight obstacles across recording sessions so that the bat could not use spatial memory for navigation, and also released the bats from different locations to avoid the development of stereotyped flight patterns. These details have been added to the revised manuscript (subsection “Experimental design”). Bat A flew for 12 sessions, and Bat B flew for 15 sessions. Figure 3—figure supplement 1 shows the bat’s flight trajectories in a single session and illustrates room coverage. Coverage was restricted in elevation, due to the height of the flight room, with a floor to ceiling dimension of approximately 250 cm. In the revision, we have elaborated on these methods to demonstrate the bats’ flight paths and coverage of the room.

3) How were the animals trained on this task and how well trained were the bats? Is it possible that the sharpening of the neural spatial resolution is dependent on how well a task is known, meaning a high expectancy of animal? Did the authors observe any changes in the sharpening on dynamic tuning of the neural activity during the days of poorer vs. better performance?

We have expanded the Materials and methods section of our manuscript to give a more detailed description of the behavioral task performed by the animals in our study. Briefly, two bats flew freely in a large experimental test room (6 x 6 x 2.5 m) and were fed mealworms throughout recording sessions to keep them active and motivated to fly, but the animals were not specifically rewarded for the task. The room was illuminated with long-wavelength lighting to preclude the bat’s use of vision, and the animals were released from different locations at the start of each trial to avoid the use of spatial memory for navigation. The obstacles the bats encountered in flight were four plastic cylinders (hard plastic as to be acoustically reflective), approximately 13 cm in diameter and 30 cm in length, and their positions were changed after each recording session. As mentioned previously, we also released the bats from different directions on each trial to emphasize the bats’ use of echolocation, rather than spatial memory. We therefore cannot examine our data from the perspective of ‘expectancy’ or ‘performance.’

4) What was the reward contingency? Neural activity in SC is believed to be modulated by upcoming reward (see for example Ikeda and Hikosaka, Neuron, 2003). Did the authors observe different patterns of neural activity near rewarded (for example landing platform) vs. non-rewarded (for instance hanging obstacle)? Was the location of the obstacles changed between recording sessions and did this manipulation had an effect on the animal behavior and neural activity?

The bats were not explicitly rewarded in our task; they were just fed after each trial (and every trial, regardless of the animal’s behavior) to keep them active and motivated to fly. We therefore cannot separate the data into rewarded vs. non-rewarded sections of the room, or based upon reward contingency. We have included these details in the revised manuscript (subsection “Experimental design”, first paragraph). As mentioned above, the location of objects was changed for each recording session.

5) Can the authors show that the bats used echolocation rather than vision to perform the task? I assume the room was dark during the experiment but how dark was it (lux levels)? Can the authors show that the bat behavior was modulated by the echo-pulses or alternatively, that they did not use vision for attending to distal cues and echolocation for attending to more proximal cues (which might provide an alternative explanation for neural responses)? This information is important in order assess the extent to which these bats used different sensory modalities (audition vs. vision) in performing different aspects of this task. As a side note, most often high-speed video tracking systems employ infrared-red lighting. Is there data for Eptesicus fuscus's wavelength sensitivity?

We thank the reviewer for the comment and agree that details of the room lighting are important to mention. These have now been added to the manuscript. Briefly, the room lighting was outside the bat’s visible range: ERG data show that *Eptesicus* does not see wavelengths longer than 600 nanometers (Hope and Bhatnagar, 1979). Our infrared cameras operate at a wavelength ~850 nanometers. These details are now included in the manuscript (subsection “Experimental design”).

6) Can the authors provide a more detailed description as to the type/size/shape of the objects used in their task? The results seem to show that the bats only needed a glimpse of the object and used this mainly to detect the objects' spatial positions, not needing more information of the spatial extent of an object. Was this indeed the case for most objects presented or were some inspected for longer durations of time? If so, did this influence the echolocation signals and corresponding neural activity?

As noted above, the obstacles were hard plastic cylinders that returned strong echoes to the free-flying bat. They were 13 cm in diameter, and 30 inches in length. Detailed descriptions of the obstacles are included in the revised manuscript. The animals did ‘attend’ to objects at different levels, as indicated by the production of SSG’s, and as our manuscript details, this increased spatial attention altered the tuning profiles of the neurons in our study.

7) What echolocation signals are actually being analyzed here? Is it only to returning echoes or also to "missed" echoes, meaning did the authors also look at all neural responses after an echolocation call was made independent of whether this resulted in a reflection or not?

In our experimental setup, all sonar calls resulted in echo returns. Spatial tuning profiles were computed for all neural responses within a trial. To further specify, for every call a bat produced, echo timings and directions were computed using the echo model and thus there would not be a case with ‘missed echoes’. The bat always received echoes each time it vocalized, either from a flight obstacle, the platform, the floor, ceiling, or a wall, which were used to construct spatial tuning profiles of sensory neurons. If a neuron showed a response, this entered into the calculation of spike probability for that location in space, whether there was a recorded echo or not. We will include these details in the revised manuscript. We have added this information to the Materials and methods subsection “Classification of neurons into sensory, sensorimotor and vocal-premotor cells”.

In addition to the analysis of neural data with respect to behavior noted above, further basic information should be analyzed and provided by authors to better assess the quality of the presented data and its validity for the authors interpretation. For instance:1) What was the quality of the sorting? How well were the clusters separated into single units? The authors should provide quantitative measures for the quality of their sorting (such as isolation index and L_ratio_).

We used a wavelet-based clustering developed by Quiroga et al., 2004 to perform the sorting. This program uses a Monte Carlo simulation to establish significance in the separation of clusters and does not report cluster separation values (like L_ratio_ or isolation index). In case the reviewers request a cluster isolation value, we would be happy to compute these and provide the values separately.

2) Furthermore, the authors describe a threshold of 200 spikes for a cell to be included in the analysis. How was this threshold chosen and does this number only include spikes recorded during flight? The authors further set a seemingly arbitrary threshold of at least 20 spikes during SSG and non-SSG which results in a total sample of 20 neurons for this entire analysis? If this is indeed the case it seems rather concerning basing conclusions on 20 action potentials and such a low number of neurons. The authors should either increase their neuron count or show statistically the strength of their threshold as valid for avoiding low sample biases for this analysis.

In our revision, we have added neurons to our data set (as explained above), provided details on how the thresholds were chosen, and added statistics to show that our results are not a result of a low sample bias. Briefly, the wavelet clustering requires a minimum cluster size threshold for isolating and separating clusters. Due to the sparse nature of firing in the bat auditory system we used a threshold of a minimum 50 spikes per cluster.

Further, we have removed the threshold of 20 spikes used for the SSG and non-SSG analysis. And now the SSG and non-SSG analysis includes all 56 distance tuned neurons. These details are now provided in the revised manuscript (subsection “Adaptive sonar behavior modulates 3D spatial receptive fields”).

3) How were the motion artifacts shown by the authors in Figure 1—figure supplement 1 characterized and corrected? How were these artifacts detected and were they constrained to more specific spatial locations (such as the landing platform) or equally distributed throughout the flight paths? Could these artifacts obscure the detection of action potentials in certain parts of the environment more than in others? What did the authors do to address the potential influence of such artifacts on the neural signals and especially for on LFP signal? For instance, motion artifacts can cause the distortion of electrical signals in low frequency bands. To what extent were gamma oscillations influenced by such interferences? If any patterns of wing motions were associated with SSGs then this could account for stronger fluctuations in LFP hence seemingly increasing the gamma power.

The wavelet based sorting method developed by Quiroga et al., 2004 can also be used to exclude motion artifacts from the neural recordings. In our experiments, the neural recordings from free-flying bats rarely showed motion artifact. As the reviewer speculates, there are some motion artifacts during landing, but we have excluded these time points from the analysis. Please see text (subsection “Surgical Procedure, neural recordings and spike sorting”, last paragraph) for further details.

The reviewer is correct in pointing out that wing beat artifacts could indeed influence lower frequency oscillations included in the LFP. We also examined whether movement artifact from the bats’ wing beats could have corrupted the LFP analysis. The bat’s wingbeat is approximately 12 Hz, whereas the frequency range for the gamma band we analyzed was 40-140 Hz. The 3^rd^ harmonic of the wingbeat, which would be in the frequency range of the gamma band, was significantly attenuated. To further ensure that movement artifact did not corrupt the analysis of the LFP, we chose channels where the power ratio between the low frequency band (10-20 Hz) and the gamma band was less than 6 dB. We identified 21 such low noise, channels (see Figure 6—figure supplement 2), which were then used for further analysis. We have added this information about the possible influence of wing beat artifacts on the LFP analysis to the revised manuscript (subsection “Local field potential”).

4) What part of the frequency band of gamma was analyzed? The authors define a very wide band (40-140Hz) but there is a clear distinction between low and high gamma frequency bands that should also be analyzed. Furthermore, did the authors observe differences both at the gamma band effect as a function of the recorded layer in the superior colliculus? (see for example the paper by Ghose, Maier, Nidiffer and Walace, Multisensory response modulation in the superficial layers of the superior colliculus, Journal of Neuroscience, 2014).

We analyzed the gamma band between 40 and 140 Hz, as explained in the methods. The reviewer’s comment prompted us to now look at the gamma band effect as a function of the dorsal-ventral axis in the SC, and we do not see any change as a function of recording depth. These details are included in our revision (subsection “Gamma power increases during epochs of sonar sound group production”, first paragraph, also Figure 6—figure supplement 1).

5) Did the selectivity of the neurons change as a function of anatomical location of the recorded neurons within the SC (such as more superficial vs. deeper layers of SC)?

Our analysis does not show any change in neural selectivity as a function of anatomic location (depth of recording location). This has been included in the revision (subsection “3D spatial tuning of single neurons in the SC of free flying bats”, last paragraph and Figure 4—figure supplement 2).

6) Stability: Figure 1—figure supplement 1 the authors show some action potentials from a single session yet this does not address in a meaningful way the question of whether the neural features observed by the authors represent a stable phenomenon related to the bat echolocation alone or is it actually modulated by different factors which are changing during the task. For such analysis, the authors should compute the stability of the neural responses of all the analyzed neurons during equally spaced portions of each session and show quantitatively that the response profiles (such as tuning curves) remain stable throughout.

We agree with the reviewer’s comments and we show, in our revised manuscript, that the spatial tuning of cells is stable across each recording session. In this context, we would like to mention that the sparseness in neural activity in the auditory system of big brown bats (Dear et al., 1993; Valentine and Moss, 1997) limits the spiking data we can use for this assessment on a per trial basis. We compared response profiles of neurons in the first and second half of recording sessions, and found that the vast majority of neurons (37 out of 46 neurons) had stable spatial receptive fields between the two halves of each recording session. These details are included in the fourth paragraph of the subsection “Construction of 3D spatial response profiles”, also, Figure 4—figure supplement 1).

7) The definition of neural selectivity in unclear. What is the formal definition by which a neuron was determined to be selective along a particular dimension? Can the authors provide a formula and statistical description for how a neuron was selected for the analysis (the paper is primarily relying on just 41 sensory selective neurons)? How were sensory, sensorimotor and vocal-premotor neurons classified? How many were recorded from each bat? What statistical threshold was used to determine selectivity? Furthermore, was the selectivity of a neuron analyzed along any orthogonal set of dimensions which are different from the canonical x, y and z dimensions?

The reviewers have made the important point that the description of how sensory, sensorimotor and vocal-premotor neurons were classified is lacking in the original manuscript. We didn’t present information on sensorimotor and vocal premotor neurons in the original version of the manuscript, because we are preparing a separate paper that focuses on these classes of neurons in the bat SC. We provide a brief explanation below, and we have added a section, in the manuscript text, detailing the methodology of the classification (subsection “Classification of neurons into sensory, sensorimotor and vocal-premotor cells”).

In the case of a passively listening animal, it is straightforward to identify and characterize sensory activity. In head restrained animals, traditionally, sensory, sensorimotor and premotor cells are identified by separating the sensory and motor behaviors in time. This allows the experimenter to solve the problem of assigning neural activity to independent behavioral/sensory events. In a freely moving animal that is interacting with physical objects in its environment, there are challenges to analyzing neural activity in this way. To address this challenge, we developed an algorithm based on the firing latency distributions of spike times with respect to echo arrival time, previous call production time, and next call production time. This algorithm classifies neurons as sensory, sensorimotor and vocal premotor, based upon the temporal relationship between echo time and spike latency.

Once a neuron was identified as sensory (see above explanation), direction information from the echo model was converted into egocentric coordinates of the bat’s instantaneous position and the X, Y and Z information was converted into azimuth, elevation and range coordinates. After normalizing the neural responses based on the amount of coverage in each of these dimensions, we fit normal curves to these responses. We also performed an ANOVA to determine the significance of spatial tuning along each dimension. These details have been added to the manuscript (subsection “Construction of 3D spatial response profiles”).

Neural selectivity was analyzed only with respect to the X, Y, and Z dimensions. Since the bat’s own calls evoked echoes that stimulated SC neurons, we could not systematically analyze responses to other stimulus dimensions, such as sound frequency or intensity.

8) The authors argue that neuron's best delay (target distance) is shifted to shorter delays (closer objects) when the bat is engaged in the production of SSGs, suggesting that distance tuning is dynamically remapped when the bat actively inspects objects in its environment. So, what actually drives the observed change in neural activity? Is it the result of higher echolocation rates? Is it that the bat simply in a different behavioral mode? Is it the physical proximity of objects? A detailed analysis of the neural responses of single neurons during SSGs across different locations in the room, either near or further away from the target and at different echolocation rates would be important for accurately assessing the observed correlated changes in neural activity.

To examine this issue, we took pulse intervals of all SSG calls and partitioned them into low and high pulse interval (PI) classes. We then computed range tuning of neurons for echoes returning from SSGs produced at different call intervals and determined whether spatial responses were remapped or sharper. We do not find any significant difference in the range tuning between the low and high PI SSG groups. In other words, the remapping and sharpening observed for SSGs does not seem to be a result of higher echolocation rates but rather the bat’s behavioral mode.

Lastly, while the animal model and experimental preparation are certainly exciting, the paper in its current form does not provide revolutionary new insight nor a methodological innovation beyond what has already been reported in previous studies. Neural recordings have already been done in freely behaving and flying bats and the dynamic encoding of spatial locations and attention by both single neurons and LFP (gamma oscillations in this case) have already been reported in a wide range of species. Yet, this paper does provide an important confirmation of previous hypothesis and data in the freely flying bat and for that it does provide important insight.In detail, the fact that perceptual sensitivity of SC neurons changes in relation to spatial cues is not novel has already been shown previously in behaving animals (See for example Lovejoy and Krauzlis, 2017). Albeit in some cases in non-human primates the head is restrained, they are performing active sensing using their eyes which are freely moving. The authors further argue that most studies have studied only 2D spatial cues but the three-dimensional tuning properties of neurons in SC have also been shown previously in the same bat species in the same lab (Valentine and Moss, 1997), albeit this was not done in freely flying bats. A dynamic modulation of neural activity with respect to the bat echolocation signals have also been previously demonstrated both during behavior in the same species (Ulanovsky and Moss, Hippocampal, 2011) as well as during flight in a different species (Geva et al., Nature neuroscience, 2016). Yet these recordings were done in the hippocampus and not in the SC as in the present study. Hence, the importance of this work is that it brings many of these components together in the superior colliculus of the flying bat. This aspect is important as it allows addressing long standing hypotheses about the tuning properties of such neurons in echolocating bats. It is important however that the authors properly and clearly delineate the statements of novelty in this study. As a side note: The authors further make the statement in the Introduction that "past studies of the SC have largely focused on sensorimotor representation in restrained animals, leaving gaps in our knowledge about the influence of action and attention on sensory responses in freely moving animals". Yet this statement is also not entirely accurate. Studies in rodents have also provided important contribution on the role of sensory input for modulating action and neural activity in the SC and should be acknowledged (see for example Felsen and Mainen, 2008: Neural substrate of sensory-guided locomotor decision in the rat superior colliculus).

We reiterate below why our study is novel and warrants publication in *eLife*:

1) 3D sensory responses in a freely behaving bat to self-generated echoes from physical objects have never been previously demonstrated.

2) While reports on 3D place fields in the hippocampus of the Egyptian fruit bat have provided an important advance in neuroscience, the hippocampus is implicated in allocentric space and memory-based vectorial representation. Here we report on an entirely different finding: 3D egocentric sensory responses. Our results are the first to show that 3D sensory space is represented in the brain of an animal interacting with objects in its physical environment, and importantly, that sensory representations are modified by adaptive vocal-motor behaviors.

3) Changes in hippocampal place field tuning at varying time intervals following sonar emissions were reported for crawling bats (Ulanovsky and Moss, 2011); however, this result does not bear on the novelty of the findings reported in our manuscript for two reasons: 1) The dynamics of hippocampal place cell tuning following sonar emissions were hypothesized to relate to echo processing time, but this was never empirically demonstrated in the 2011 paper. No sonar echoes were recorded, analyzed or computed from spatial coordinates, and 2) The echolocation behavior of the crawling bat did not show the natural call intervals or sonar sound groups of the free-flying bats. In particular, the call intervals used to analyze the hippocampal place field tuning in the Ulanovsky and Moss, 2011 study were 0-78 ms, 78-210 ms, 210-540 ms, and >540 ms, to equate the number of spikes used to construct place fields over different time periods following sonar calls. It is important to note that these intervals are far greater than those produced by flying bats engaged in spatial navigation and provide no indication of the animal’s behavioral state. Indeed, the call intervals produced by the bats in our current study ranged between 8 and 80 ms, falling almost entirely within the shortest time window used in the hippocampal place field tuning analysis.

4) We assert that the remapping of 3D echo-evoked responses in the midbrain SC of the echolocating big brown bat is entirely new and stands apart from Geva et al.’s 2016 report of hippocampal place field remapping in Egyptian fruit bats exposed to different sensory environments (vision vs. echolocation). Not only do the data from the two studies come from recordings in different species and different brain structures, but also in experiments with different dependent and independent variables: Geva et al. showed remapping of hippocampal place fields in Egyptian fruit bats tested in two distinct environments, while we report here remapping of sensory response profiles in the SC of big brown bats that adapted echolocation behavior to inspect objects within a single test environment. It is also worth noting that the Egyptian fruit bats are highly visual (in contrast to the big brown bat), produce tongue clicks for echolocation, not laryngeal calls, and this animal does *not* dynamically modulate its sonar signal design as it inspects objects in its surroundings. Therefore the sensory remapping results we report in this manuscript could not be obtained from the Egyptian fruit bat.

5) While past studies have implicated the SC in spatial attention and perception through local inactivation (Lovejoy and Krauzlis, 2017; McPeek and Keller, 2004), our work differs in that we recorded activity in single SC neurons in freely behaving animals and demonstrate remapping and sharpening of 3D neurons with changes in the animal’s echolocation behavior.

6) The characterization of 3D response profiles in freely echolocating bats is an important breakthrough. For many decades, scientists have mimicked natural echolocation in restrained (often anesthetized) passively listening bats, without testing the validity of this approach. Would a vision scientist not consider it critical to compare neural responses to visual stimuli in paralyzed animals viewing moving patterns with a behaving animal moving its eyes to scan a stimulus? Here, for the *first* time, we present data that not only demonstrates that auditory neurons in flying bats show 3D auditory spatial tuning, but also the discoveries that 1) tuning is sharper than reports from passively listening bats and 2) tuning is modulated by the bat’s echolocation behavior.

7) We thank the reviewer in drawing our attention to the work by Felsen and Mainen. We now discuss their work in the manuscript (Introduction, fourth paragraph).

8) Importantly, we believe that our research can inspire colleagues to conduct related studies in other species, which would advance a more complete understanding of nervous system function in the context of real-world, natural behaviors.

[Editors' note: further revisions were requested prior to acceptance, as described below.]

The manuscript has been improved but there are some remaining issues that need to be addressed before acceptance, as outlined below, and with more details provided in the individual reviews. All of these points relate to the analysis and presentation of the data, and the reviewers agreed that the novelty and impact of this study will be clearer once this is done.1) Quantification of the behavior (represented either by correlation of flight trajectories, heatmaps or a different method).

The question raised by the reviewers pertains to how much spatial memory is guiding the bat’s flight behavior. In the newly revised manuscript, we report quantitative analyses of the amount of stereotypy in the bats’ flight behavior, using an approach similar to Barchi et al. (2013) and Falk et al. (2014). Specifically, we performed 2D spatial cross correlations of occupancy histograms for every trial within a recording session. Using this technique, high correlation values indicate stereotyped flight paths from trial to trial, and these have been interpreted as an indication of the bat’s use of spatial memory (Barchi et al., 2013). Similarly, low correlation numbers have been interpreted as use of active sensing rather than spatial memory (Falk et al., 2014).

In our study, bats were released from different locations in the room on each trial, and our analysis shows (Figure 1—figure supplement 1) low correlation values in flight paths across trials. Based on this finding, we argue that the bats in our study relied on sensory input to guide their flight trajectories, and not spatial memory. We believe that a correlation analysis of flight paths is more appropriate than a heatmap to demonstrate this point. A heatmap only provides *coverage* information, and would therefore be insufficient to describe how stereotyped flights are from trial-to-trial as an indication of the use of spatial memory in navigation. In our experiment, *coverage* is important to the analysis of single neuron spatial tuning, and for this analysis, we show coverage across azimuth, elevation, and distance in Figure 3—figure supplement 1. This analysis has been included in the main manuscript (Results, first paragraph and subsection “Analysis of flight behavior”).

2) Since there are data from only two bats, perhaps the authors could show the distribution of the main results across those two bats (as one would do for non-human primates which often have two subjects) to allow the readers to compare the responses.

We agree that it is important to show the separate results for the two bats in our study to allow the reader to assess the consistency of the results between animals. In the revised manuscript, we have color coded the summary data in Figure 4 and Figure 5 (Bat 1 in green, Bat 2 in brown), and parsed the data in Figure 6 by individual bat.

3) Provide the standard measurement of unit-isolation quality by showing the distribution of L_ratio_ and isolation distance for all of the analyzed neurons and exclude neurons which are clearly multi-unit. The concern is that biases can emerge from noisy analysis relying on very low spike counts. The authors should provide a measure of what is a reasonable threshold of number of spikes for a single neuron such that it can be safely included in the different analysis, or show that their results do not depend on spike counts.

We have now provided two different measures of unit-isolation quality in the revised manuscript. These are measures that are typically employed to measure cluster separation of tetrode data: L_ratio_ and isolation distance (Schmitzer-Torbert, et al., 2005; Saleem, et al., 2013). In prior studies, L_ratio_’s less than 0.07 and isolation distances greater than 15 were used as criteria for well separated clusters. In our data, all L_ratio_’s were less than 0.05, and isolation distances were greater than 15. These details have been added to the manuscript (Results, third paragraph; subsection “Surgical Procedure, neural recordings and spike sorting”, third paragraph and Figure 1—figure supplement 3).

Reviewer #1:The authors have been very responsive to my previous review. I accept their reasons for relying on their model for the timing of echo-evoked spikes.With respect to neurons recorded, although data are still from only 2 bats, they have increased the number of neurons analyzed. The authors now report 182 single neurons recorded in the SC, with 67 being selective to pulse-echo pairs. The authors have answered questions about unit isolation with a citation to the Quiroga et al. 2004 paper, and also point out that data from both bats show similar results. More information should be provided about these units. The authors could provide data on their units to support divisions into sensory, motor, etc., including latency and rate, as well as spike sorting criteria.

In the manuscript, we provide the criteria used to classify units into sensory, sensorimotor and vocal-premotor units in the subsection "Classification of neurons into sensory, sensorimotor and vocal-premotor cells”. The mean response latencies of single sensory neurons we recorded is 5.9 ± 3.4 ms. In more detail, the minimum spike latency was 3 ms and the minimum s.d. of latency was 1 ms. The median s.d. of the response latencies for the 67 sensory neurons was 3.8 ms. Previous publications have reported a wide range of response latencies in SC neurons of the passively listening bat, as long as 40 ms, but also as short as 4 ms (Valentine and Moss, 1997), 3.6 ms (Jen et al., 1984) and 4 ms (Wong, 1984), and short latency responses are likely mediated through a direct projection from the nucleus of the central acoustic tract to the SC (Casseday et al., 1989). These results have been included in the last paragraph of the subsection “Construction of spatial response profiles”.

In Author response image 1 the spike latencies are shown on the x-axis (ms) and the spike probabilities on the y-axis for typical examples of sensory neurons (panel A), vocal premotor neurons (panel B) and sensorimotor neurons (panel C). Note the negative latencies for the premotor neural activity. The blue dashed line indicates the onset of the echo (auditory stimulus) and the red dashed line indicates the onset of the vocalization. Gaussian fits of the latency data are shown in blue and red, for sensory and motor neurons, respectively.

In the revised manuscript we have also gone into more detail in regards to our wavelet-based spike sorting method. This includes new analysis examining the L_ratio_ and isolation distances of the spike sorting clusters. These are measures that are typically employed to measure cluster separation of tetrode data: L_ratio_ and isolation distance (Schmitzer-Torbert, et al., 2005; Saleem, et al., 2013). In prior studies, L_ratio_’s less than 0.07 and isolation distances greater than 15 were used as criteria for well separated clusters. In our data, all L_ratio_’s were less than 0.05, and isolation distances were greater than 15. These details have been added to the manuscript (Results, third paragraph; subsection “Surgical Procedure, neural recordings and spike sorting”, third paragraph and Figure 1—figure supplement 3).

In the revision, the authors provide more data on their SSG responses. It would be helpful to find out how the 26 units shown for the SSGs were classified? The only group of neurons that I can find with n=26 are the vocal premotor group. Are these the same units?

We have revised the manuscript (subsection “Adaptive sonar behavior modulates 3D spatial receptive fields”, last paragraph) to clarify how responses to SSGs were classified and the numbers of neurons in each category. We apologize for the confusion in our previous version.

Regarding the second question, we reported that 26 vocal premotor neurons were characterized (subsection “3D spatial tuning of single neurons in the SC of free flying bats”, second paragraph); however, these were different from the 26 neurons in the SSG/gamma power analysis.

For the gamma power (SSG v/s nonSSG) analysis we analyzed all the neurons, which showed significant tuning in range (n = 56). Out of these neurons, we were only able to perform the analysis on 26 neurons (from 21 different channels), which were not affected by the bat’s wing beat artifact. These details have been added to the text in the Materials and methods subsection “Local field potential.”

The other revisions greatly add to the paper, providing data on bat flight paths, and number of sessions in which responses were recorded.

Thank you!

Reviewer #3:While I appreciate the response from the authors I still find the information provided lacking on the three main domains I described in my original review:1) Behavioral data analysis and presentation: Still there is no detailed description and importantly, quantification of the behavior. The authors say that bats were released from different locations and did not exhibit stereotyped behaviors but as requested, they should show this. As requested previously, can the authors provide a quantitative assessment of the reproducibility (or lack-thereof) of both the starting positions and flight trajectories. This can, for example, be done in the form of a heat-map illustration the spatial distribution (in 3D space) of the flight trajectories. But I encourage the author to provide a different assessment of this important point as well such that their argument for more variable flight trajectories and starting positions are better supported.

We appreciate this reviewer’s concern and have followed up with a quantification of the bat’s flight trajectories, which shows that the bat’s flight behavior in this study was not stereotyped. In the revised manuscript, we have provided a quantification of the trial-to-trial correlations in flight trajectory. As mentioned above, in the newly revised manuscript, we chose to quantitatively measure the amount of stereotypy in the bats’ flight behavior, following a method used by Barchi et al. (2013) and Falk et al. (2014). In this analysis, we performed 2D spatial cross correlations of occupancy histograms for every trial within a recording session. Using this technique, high correlation values indicate stereotyped flight paths from trial to trial and these have been interpreted as an indication of the bat’s use of spatial memory (Barchi et al., 2013). Conversely, low correlation numbers have been interpreted as use of active sensing rather than spatial memory (Falk et al., 2014).

The detailed methodology is as follows. Occupancy histograms were created by collapsing the 3D trajectory data to 2D plan projection (*x,y and x,z*). The number of points across a set of flight paths that fell inside 10 cm^2^ bins was counted. These points were converted to probabilities by dividing each bin count by the total number of points across each set of flights. After normalization, the occupancy histograms of trials could be compared within each session. The next step was to compute the autocorrelation of each trial and cross correlation of each trial with every other trial. The maximum value of each 2D cross correlation was divided by the maximum value of the autocorrelation. This ratio is shown as a matrix for a representative session for both bats in Figure 1—figure supplement 1. The value of each square along the diagonal is one (yellow on the color bar), as it represents the autocorrelation of each flight trajectory. Cooler colors indicate minimum correlation between flight trajectories and warmer colors indicate stereotypy between trajectories. Further, we used 8 positions (a-h) for releasing Bat 1 and 6 positions (a-f) for releasing Bat 2. These are indicated on each plot to better allow evaluation of stereotypy when the bat was released from the same release point.

In our analysis (Figure 1—figure supplement 1), we found a very low correlation in flight paths across trials, and argue that this analysis, along with the bat’s adaptive sonar behaviors, provide evidence that the bats did not rely on spatial memory to guide their flight. We believe that a correlation analysis of flight paths is more appropriate than an occupancy heatmap to demonstrate this point. This analysis has been included in the main manuscript (Results, first paragraph and subsection “Analysis of flight behaviour”). The question raised by the reviewer pertains to whether spatial memory may have been guiding the bat’s flight behavior. An occupancy heatmap only provides *coverage* information, and would therefore be insufficient to describe how stereotyped flights are from trial-to-trial. In our experiment, *coverage* is important to the analysis of spatial tuning, and for this analysis, we show coverage across azimuth, elevation, and distance in Figure 3—figure supplement 1.

Also, the information on reward contingency is lacking. How many trials in a session resulted in landing on a platform and how many did not? When the bats did not land on the platform were they also rewarded? Much more details of the behavior are required and at the moment not provided by the authors. Again, this information is important for addressing the nature of the neural responses as postulated by the authors and align their results with findings from other species (such as primates, rodents, etc…). This will allow the authors to engage a broader audience beyond the single species of bats.

In this study, we did not impose a ‘reward contingency’ upon the bat’s performance. We fed the bats mealworms at the end of each trial to keep them active, regardless of where they landed. Bat 1 landed on the platform (70% of trials) and elsewhere in the room (30% of trials), and was fed each time it landed. Bat 2’s task was simpler, and it flew around our experimental test room and avoided crashing into hanging obstacles. When Bat 2 landed on the wall, which marked the end of a trial, it was fed. We have changed the text to emphasize that the animals weren’t conditionally rewarded for different behaviors, but merely fed during the course of the experiment (Results, first paragraph and in the subsection “Experimental design”).

2) Analysis of neural data: Despite the request in the previous round of reviews there is no quantification of the quality of the neural signal. As requested, the authors should provide some assessment of the quality of their neural data in the form of isolation indexes and L_ratio_. Again, such measures are fairly standard in neural analysis and would allow the readers to assess the data more properly. Also, the authors have now removed the threshold on the minimal number of action potentials for a neuron to be included in their analysis. This allowed them to increase the N of "valid neurons" without a quantification of the dependence of the sensitivity of the observed tuning curves on the number of action potentials included in the analysis. Furthermore, this was not requested by the reviewers and bring about concerns regarding conclusions being made based on very low number of spikes and without support that this cannot bias the result. I encourage the authors to reinstate the threshold and importantly, provide a quantitative threshold that would assure the results are not biased by low numbers of action potentials included in the analysis.

Quantification of neural signals and clustering:

We apologize for not including these points in earlier versions of our manuscript. In the revised manuscript, we have provided information on the L_ratio_ and isolation distance for all wavelet clustered data. The clusters are well within the ranges used in prior work (Schmitzer-Torbert, et al., 2005; Saleem, et al., 2013) for significantly separated clusters (L_ratio_ < 0.07, isolation distance > 15). We have added this information to the third paragraph of the Results and to the third paragraph of the subsection “Surgical Procedure, neural recordings and spike sorting”, and provided a figure for the L_ratio_ analysis (Figure 1—figure supplement 3).

Power analysis replaced the threshold criterion for SSG v/s nonSSG analysis:

As the reviewer notes, we removed the threshold criterion, which was included in the first submission. However, we would like to clarify that the increase in total number of units now included in our paper was due to the addition of more data sessions. New video analysis tools permitted flight trajectory reconstructions in some trials that were originally not analyzed, due to poor video quality. In the original submission of the paper, which reported on the analysis of 20 range-tuned neurons, 17 showed a significant effect of echolocation behavior on range tuning. In the first revision of the manuscript, in which we added more neurons, we also removed the threshold criterion, and the number of units increased to 56. In response to the reviewer’s concern about removing the threshold criterion, we have now in the second revision, adopted a more rigorous power analysis to determine which units should be included in our data set (see subsection “Adaptive sonar behavior modulates 3D spatial receptive fields” and subsection “Power analysis of sample sizes for the SSG and non-SSG spatial tuning comparisons”), and we report on an n of 53 neurons in Figure 5 and an n of 51 in Figure 5.

For the SSG and non-SSG analysis we separated spiking activity when the bat produced SSGs and nonSSGs. This resulted in some of the groups having a low spike count. To ensure that for each comparison, and for each neuron, we had enough statistical power to reliably analyze the data, we performed a permutation test. We only included the cells, which passed the test at the p < 0.05 criterion level, which excluded 3/56 cells for Figure 5/56 cells for Figure 5. We have included these additions in the manuscript (see the aforementioned subsections).

As a further confirmation of the analysis (rank sum test) used to check whether the neurons showed a significant shift in range tuning (Figure 5), we used the software G*Power (Faul, F et. al; 2009) to estimate the statistical power, given the sample sizes and standard deviation of each SSG-nonSSG group, for each and every neuron. We would like to note that the power analysis for a non-parametric test (like the rank sum test) also requires an assumption of normality. We used the Lehmann technique (available as an option in G*Power) and checked our results at α < 0.05. We went a step further and identified all neurons for which both the SSG and nonSSG groups were normally distributed (we used the Anderson Darlington test as a test of normality). For these neurons, which passed the normality test, we estimated the power (using the Matlab command *sampsizepwr*). We would like to emphasize that both the G*Power analysis and the *sampsizepwr* analysis yielded similar results.

The above sets of analyses give us confidence that our results are not biased by low spike counts and are robust.

3) Novelty: The points raised by the authors are still in agreement with the fact that this paper very elegantly puts together pieces of data that have been previously reported elsewhere, either in bats or in other species and mostly serve as a verification of previous findings. Furthermore, some of the statements made by the authors are unclear. For example, when comparing to the work of Geva et al., the authors claim that the Egyptian bat does not dynamically modulate its sonar signal in response to its surrounding but the senior author of the paper is an author on a paper stating that it does (S. Danilovich, A. Krishnan, W. J. Lee, I. Borrisov, O. Eitan, G. Kosa, C. F. Moss, Y. Yovel (2015) Bats regulate biosonar based on the availability of visual information Current Biology 25(Feng et al., 1978), 1107-1125), which is puzzling. Furthermore, other studies from Yosef Yovel and Nacham Ulanovsky have demonstrated that this species does in fact dynamically change the directionality of its sonar beam in responses to the acoustic features of its environment, and the senior author of the current paper is also an author on that manuscript: Y. Yovel, B. Falk, C. F. Moss, N. Ulanovsky, (2011) Active control of acoustic field-of-view in a biosonar system, PLoS Biology, 9(Bichot et al., 2005): e1001147. (Open article).

The data reported in our manuscript goes beyond putting together pieces of data that have been previously reported elsewhere, and we expand further on the novelty of our findings in the second revision (subsection “3D allocentric versus 3D egocentric representations in the brain”). We now recognize that some key information was not clearly conveyed in our previous response to the reviewer. 1) The fact is that the Egyptian fruit bat produces tongue clicks, not laryngeal vocalizations, and therefore. it cannot modulate the spectro-temporal features of its echolocation calls in response to echoes it gathers from the environment. As the reviewer points out, the Egyptian fruit bat has been shown to modulate the angular separation of the beam axes of sonar clicks in a pair (Yovel et al., 2011 PLoS Biology), and this finding simply illustrates that the echolocation behavior of the Egyptian fruit bat is not as primitive as previously believed. However, the Egyptian fruit bat cannot exhibit the rich dynamic sonar behavior of a laryngeal echolocator. 2) Further, none of the published studies from Nachum Ulanovksy’s lab have included quantitative analyses of the echolocation signals produced by the Egyptian fruit bat in the context of hippocampal activity. Indeed, Ulanovsky and his team have yet to consider if the timing of Egyptian fruit bat echolocation signals influences hippocampal cells. Even their recent Science paper reporting on “social place cells” does not investigate modulation of hippocampal activity by the animal’s social calls (produced by the larynx). In a 2011 paper published by Ulanovsky and Moss, place cell tuning of hippocampal neurons in the crawling big brown bat was analyzed with respect to the time elapsed following sonar emissions, but the call intervals included in this analysis were far larger than those observed in free-flying bats inspecting their environment through echolocation, and quantitative analyses of the bat’s adaptive sonar behavior in this prior study were not carried out. We assert that our study of *sensory-evoked* neural activity in the midbrain superior colliculus of the free-flying laryngeal echolocating bat is entirely novel and shares no overlap with published work on the hippocampus of the free-flying Egyptian fruit bat.

In summary, while I am generally supportive of this important work I still feel that the authors should provide more detailed responses to the requests of the referees and frame their work better with respect to the vast knowledge on the neurophysiological properties of SC neurons across species. The latter would only benefit the authors as it will allow them to extend the impact and interest in their work to a broader audience, such as the readers of eLife.

We have made every effort to respond fully to your comments and suggestions, and we thank you for taking the time to help us improve our paper.